# Language-Driven Interactive Traffic Trajectory Generation

**Junkai Xia**[1,3*]   **Chenxin Xu**[1,3*]   **Qingyao Xu**[1,3]   **Yanfeng Wang**[1,2]   **Siheng Chen**[1,2,3†]

[1]Shanghai Jiao Tong University    [2]Shanghai AI Laboratory
[3] Multi-Agent Governance & Intelligence Crew (MAGIC)

## Abstract

Realistic trajectory generation with natural language control is pivotal for advancing autonomous vehicle technology. However, previous methods focus on individual traffic participant trajectory generation, thus failing to account for the complexity of interactive traffic dynamics. In this work, we propose InteractTraj, the first language-driven traffic trajectory generator that can generate interactive traffic trajectories. InteractTraj interprets abstract trajectory descriptions into concrete formatted interaction-aware numerical codes and learns a mapping between these formatted codes and the final interactive trajectories. To interpret language descriptions, we propose a language-to-code encoder with a novel interaction-aware encoding strategy. To produce interactive traffic trajectories, we propose a code-to-trajectory decoder with interaction-aware feature aggregation that synergizes vehicle interactions with the environmental map and the vehicle moves. Extensive experiments show our method demonstrates superior performance over previous SoTA methods, offering a more realistic generation of interactive traffic trajectories with high controllability via diverse natural language commands. Our code is available at `https://github.com/X1a-jk/InteractTraj`

## 1 Introduction

Driving simulations are increasingly vital in the development of autonomous driving [1, 2, 3, 4, 5]. By projecting real-world scenarios into virtual environments, driving simulation enables the generation of driving data in diverse conditions at a significantly reduced cost, especially in safety-critical scenarios. Trajectory data, representing the driving behaviors of traffic vehicles, serves as a key part of the driving simulation. This paper focuses on the generation of traffic trajectories.

One of the most critical aspects of trajectory generation is *controllability*, which involves generating highly realistic trajectory data tailored to specific user needs. Several works have been proposed to prompt a controllable traffic trajectory generation. TrafficGen [4] enables the generation of traffic scenarios conditioned on a blank map with controllable vehicle numbers. CTG [6] allows users to control desired properties of trajectories using signal temporal logic at test time like reaching a goal or following a speed limit by a guide sampling in the diffusion process. However, given that these control signals are pre-defined, their flexibility is inherently limited.

With the rise of large language models, researchers have begun to use human natural language to achieve a more flexible and user-friendly control. A representative work is LCTGen [7], which leverages a large language model to transform text descriptions into structured representations, followed by a transformer-based decoder to generate corresponding scenarios. However, this work exclusively focuses on individual traffic participant trajectory generation, disregarding the *interactions* between multiple trajectories. Such interactions, crucial for replicating the dynamic, involve the

---

[*]These authors contributed equally to this work.
[†]Corresponding author.

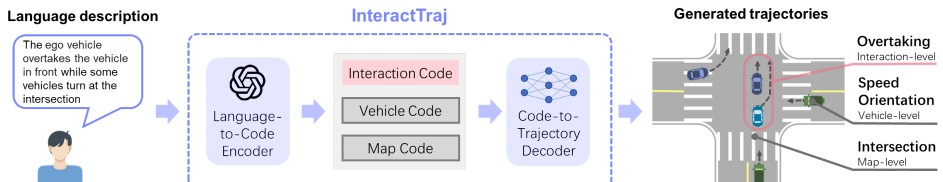

Figure 1: Overview of InteractTraj. InteractTraj uses a series of semantic interaction-aware numerical codes to depict interactive trajectories. An LLM-based language-to-code encoder converts language descriptions into numerical codes, which are then transformed into interactive trajectories by a code-to-trajectory decoder.

complex interplay between various participants' movements and decisions. The absence of interaction modeling causes limited controllability in trajectory generation. For instance, traffic jams, involving many vehicles, cannot be accurately generated. Yet, generating these scenarios is essential as they highlight the vehicles' capabilities to respond to real-world challenges.

To achieve more realistic and controllable trajectory generations, in this work, we propose InteractTraj, a novel generator that generates interactive traffic trajectories from natural language descriptions. The key design rationale of InteractTraj is to interpret abstract trajectory descriptions into concrete formatted interaction-aware numerical codes and to learn a mapping between these formatted codes and the final interactive trajectories. Specifically, InteractTraj consists of two modules: an LLM-based language-to-code encoder and a code-to-trajectory decoder. 1) To interpret user language commands, the language-to-code encoder utilizes a novel *interaction-aware encoding strategy*, which uses an LLM with interaction-aware prompts to convert language commands into three types of numerical codes, including interaction, vehicle and map codes. As the core to model interactive relationships of vehicles, the interaction codes consist of key factors, including relative position and relative distance. These factors are designed to be discrete to have semantic meanings that correspond to LLM and are formed in series to model the temporal continuity of interactions. 2) To produce interactive traffic trajectories, the code-to-trajectory decoder employs a novel two-step *interaction-aware aggregation strategy* that integrates code information. This approach synergizes vehicle interactions with environmental map data, thereby using these interactions to enrich the realism and coherence of vehicle trajectories. Compared to previous work [7], which generates independent traffic trajectories, InteractTraj is capable of generating realistic and interactive traffic trajectories with enhanced controllability through language commands.

We conduct extensive experiments on the Waymo Open Motion Dataset(WOMD) and nuPlan and show that InteractTraj can generate realistic interactive traffic trajectories with high controllability through various natural languages. Our method achieves SoTA performance with an improvement of 15.4%/18.7% on average ADE/FDE over previous methods on WOMD, and 17.1%/20.4% on average ADE/FDE on nuPlan. Our method also achieves a more realistic generation under different user commands including vehicle interactive actions of overtaking, merging, yielding and following. We conduct user studies showing our method has $47.5\%$ higher average user preference compared to the baseline method. We summarize our contributions as follows:

• We propose InteractTraj, the first language-driven traffic trajectory generator that can generate interactive traffic trajectories. The core idea of InteractTraj is to bridge abstract trajectory descriptions and generated trajectories with formatted interaction-aware numerical codes.

• We design a novel interaction interpretation mechanism with LLM in the language-to-code encoder and a two-step feature aggregation to fuse interaction information for more coherent generation in the code-to-trajectory decoder.

• We conduct extensive experiments and show that InteractTraj is capable of generating realistic interactive traffic trajectories with high controllability through various natural languages.

## 2 Related Work

### 2.1 Traffic Trajectory Generation

Traffic trajectory generation is crucial in intelligent transportation systems, producing all agents' trajectories in a scene from provided maps or historical data. Traditionally, rule-based methods [8, 1, 9, 2, 10, 3, 11] employed heuristic models to encode traffic rules like lane-keeping and following the leading vehicle but lack diversity and realism due to fixed rule patterns. Recently, learning-based methods [12, 13, 14, 15] have emerged to generate more realistic traffic trajectories by learning from

real-world data. However, these methods usually face challenges with controllability, unable to fulfill specific requirements like instructing a vehicle to turn left, and they rely on past trajectories that are expensive and difficult to obtain. There is growing interest in controllable trajectory generation [4, 6, 16, 7], focusing on customizing trajectories to meet diverse user requirements. TrafficGen [4] generates a specific number of vehicles and their trajectories on a blank map. CTG [6] uses a loss function to guide trajectory generation according to user controls. influenced by LLMs, language-driven traffic scenario generation is emerging. CTG++ [16] employs LLM to convert user queries into a loss function for realistic, controllable generation. While CTG and CTG++ require costly past trajectories, limiting their practical deployment, LCTGen [7] generates scenarios purely from language descriptions using an LLM-based interpreter and a transformer-based generator. However, these methods lack interaction awareness and struggle with complex text descriptions. Our approach addresses these issues with interaction-aware code representation and refined vehicle behavior control.

## 2.2 Motion Prediction

Motion prediction and trajectory generation are closely related concepts in the field of autonomous systems and robotics since both approaches aim to anticipate the future state of agents. Motion prediction models are often used as backbones to convert latent states into agent trajectories as part of the generated scenarios, allowing for the simulation of realistic traffic scenarios. Early methods [17, 18, 19] utilize various physics-based kinematic models for modeling agent behaviors and predicting trajectories. With the development of deep learning and neural networks, RNN and LSTM-based structures are applied for trajectory prediction [20, 21, 22, 23, 24] due to their proficiency in processing sequential data. To handle more complex trajectory prediction tasks where multiple agents are involved, models in recent years have also incorporated methods such as diffusion or transformer [25, 26, 27, 28] to achieve more accurate results. [29, 30, 31, 32, 33] achieve better results on multi-agent motion prediction tasks by focusing on interaction and relational reasoning. Trajectory generation facilitates the creation of realistic scenarios, acting as supplementary data for the development and evaluation of prediction models.

## 2.3 Large Language Models and Their Multimodal Applications

Recent years have seen dramatic advancement in the development of Large Language Models (LLMs) [34, 35, 36, 37, 38, 39, 40] such as ChatGPT [41] and GPT-4 [38]. The success of LLMs triggers a boom in multimodal tasks that require comprehensive understanding across multiple modalities, including text [42, 43, 44, 45, 46], audio [47, 48, 49, 50], motion [51, 52, 53, 54, 55] and so on. Notable examples including DALL-E [56] and Sora [57]. DALL-E [56] treats text and image tokens as a unified data stream, generating realistic images from text input. Sora [57] demonstrates the ability to create long, realistic, and imaginative videos from text descriptions. In this work, we focus on language-driven trajectory generation. Inspired by [7], our method uses GPT-4 [38] as the language encoder to leverage the deep traffic scene understanding and reasoning capabilities of LLMs. We design an interaction-aware code and prompt GPT-4 to convert language input into these codes, which contain detailed information about interactions, vehicles and map.

## 3 Problem Statement

Language-driven traffic trajectory generation aims to create realistic trajectories of traffic participants over a period of time according to language descriptions. Given a language description $L$, our goal is to propose a scenario generation model $\mathcal{G}(\cdot)$ so that the generated corresponding traffic trajectories $\mathbb{S} = \mathcal{G}(L)$ are realistic and match with the language description. Here $\mathbb{S} = [\mathbf{S}_1, \mathbf{S}_2, \ldots, \mathbf{S}_N] \in \mathbb{R}^{N \times T \times 2}$ represents the trajectory of $N$ vehicles over $T$ timesteps, where $\mathbf{S}_i = [s_i^1, s_i^2, \cdots, s_i^T] \in \mathbb{R}^{T \times 2}, \forall i \in \{1, \ldots, N\}$, and $s_i^t \in \mathbb{R}^2$ denotes the 2D positions of vehicle $i$ at the $t$-th timestep.

## 4 Methodology

### 4.1 Architecture Overview

InteractTraj is a language-guided interactive traffic trajectory generation framework that generates realistic vehicle trajectories based on natural language descriptions. The core idea of InteractTraj is to use a series of semantic numerical codes to depict interactive trajectories and learn a transformation between these codes and the interactive trajectories, see Figure 1 for a sketch. InteractTraj consists of two parts: an LLM-based language-to-code encoder and a code-to-trajectory decoder. The language-to-code encoder is designed to interpret language commands and turn the commands into three types

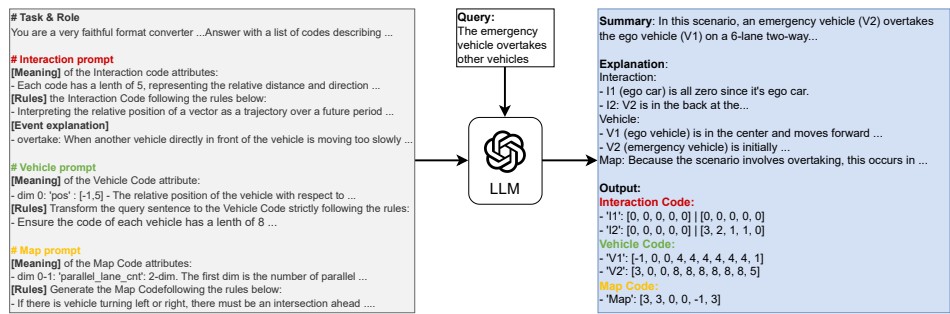

Figure 2: Sketch of interaction-aware prompt and numerical codes.

of numerical codes, including interaction codes, vehicle codes and map codes. The code-to-trajectory decoder then transforms these codes to produce interactive traffic trajectories.

Mathematically, given the language description $L$, InteractTraj generates the traffic trajectories $\widehat{\mathbb{S}}$ by

$$\mathbf{m}, \mathbf{V}, \mathbf{I} = \mathcal{E}(L), \quad \widehat{\mathbb{S}} = \mathcal{D}(\mathbf{m}, \mathbf{V}, \mathbf{I}), \tag{1}$$

where $\mathcal{E}(\cdot)$ represents the language-to-code encoder and $\mathcal{D}(\cdot)$ represents the code-to-trajectory decoder. $\mathbf{m}$ is the map codes representing the environment map information, $\mathbf{V}$ is the vehicle codes representing the vehicle's individual driving information and $\mathbf{I}$ is the interaction codes representing the vehicle interaction information. We illustrate the detailed structures of these codes in the following.

## 4.2 Language-to-Code Encoder

The language-to-code module, $\mathcal{E}(\cdot)$, distills essential information from input natural language descriptions and transforms this information into interaction-aware numerical codes. This transformation leverages large language models, such as GPT-4 [38]. The whole language-to-code encoder incorporates two key designs: the structure of the interaction-aware numerical codes and the tailored prompts for the large language model. The numerical codes are comprised of three components: interaction codes, vehicle codes, and map codes. To depict vehicle interactions concretely, we design the format of the interaction codes with the relative factors modeling. To interpret the abstract interaction-aware descriptions into the code format, we design prompts with interaction descriptions to assist the LLM.

**Interaction codes I.** Interaction codes encapsulate vehicle interactive relationships. The core idea is that spatial relationships and changes between vehicles significantly affect their perception and reactions, revealing their interactions. To capture high-level actions and interaction tendencies, we resample the vehicle attributes at regular intervals across $T$ timesteps. We denote $\mathcal{T}$ as the set of timesteps of the resampling process. To effectively model these interactive relationships, the designed interaction codes consist of two key factors, relative distance and relative direction, motivated by the representation of polar coordinates. Formally, the interaction codes are denoted by $\mathbf{I} = [(p_j^t, d_j^t)]_{j \in \{1,...,N\}, t \in \mathcal{T}}$, where $p_j^t / d_j^t$ is the relative direction/distance of $j$th vehicle with the ego interacted vehicle at the $t$-th sampling timestep. To enrich the interaction code with semantic meanings, we discretize the relative direction/distance. Specifically, we divide the surrounding space centered by ego vehicle into six regions: front, rear, left front, left rear, right front and right rear. The relative direction $p_j^t$ of agent $j$ can thus be represented by the index of the region in which agent $j$ is located at time $t$. The relative distance $d_j^t$ is also discretized by dividing with a fixed interval and then rounding down. This discrete code representation with semantic meanings facilitates the use of LLM to link code values with corresponding language commands.

**Vehicle codes V.** The vehicle codes contain the information of vehicle individual driving states. To describe the vehicle driving states, the vehicle codes consist of two components on different trajectory scales: the global trend and the detailed movement. Formally, the vehicle codes are denoted by $\mathbf{V} = [r_i; \mathbf{a}_i]_{i \in \{1,...,N\}}$, where $r_i$ is the trajectory type of agent $i$ modeling trajectory global trend and $\mathbf{a}_i$ is the vehicle states of agent $i$ modeling detailed movement. Specifically, we categorize the trajectory type into stop, straight ahead, left turn, right turn, left change lane, and right change lane. $r_i$ is the category index. Trajectory states $\mathbf{a}_i = [o_i, q_i, [v_i^t]_{t \in \mathcal{T}}]$, contain the initial orientation $o_i$, initial position $q_i$, and the discrete speeds $[v_i^t]_{t \in \mathcal{T}}$ of agent $i$ at sampled timesteps.

**Map codes m.** The map codes contain the information on key map features. We adopt the representation $\mathbf{m} \in \mathbb{Z}^6$ similar to [7], which represents the number of lanes in each of the four

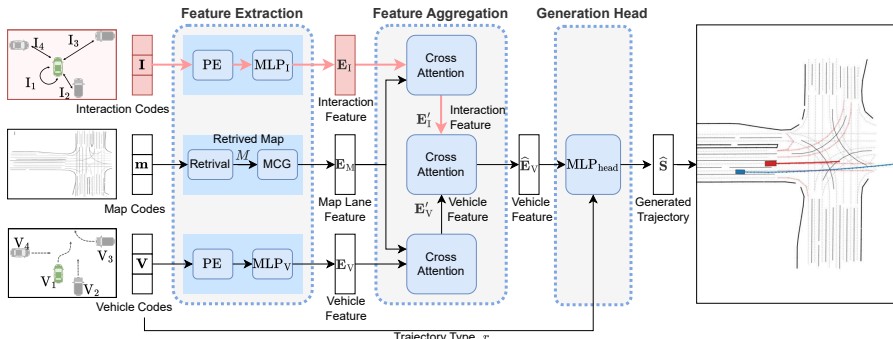

Figure 3: The architecture of code-to-trajectory decoder. The decoder generates vehicle trajectories by fusing and decoding information between vehicles and interactions.

directions, the distance between ego vehicle and the intersection, and the lane index of the ego vehicle, respectively.

**LLM prompts.** Given a language description $L$, we utilize carefully designed interaction-aware prompts to help the large language model analyze the descriptions and extract interact-related information, enabling it to generate corresponding interaction-aware numerical codes that align with the language description. The prompt mainly incorporates three key components: 1) `[interaction prompt]` interaction prompt defines the format of interaction code, explains some key interaction events to help with the better understanding of the scenarios and informs LLM to interpret the possible interaction behaviors inferred in the descriptions and output the corresponding interaction codes $I$ of all the vehicles involved by analyzing the vehicles' relative distance and position relationships. 2) `[vehicle prompt]` vehicle prompt defines the format of vehicle code and explains driving rules to make the generated scenarios more realistic, such as the need to slow down when turning, etc. 3) `[map prompt]` map prompt defines the format of map code and make LLM analyze whether intersections or roundabouts are involved and decides the number of lanes of all directions according to the number of vehicles and their orientations to fill out the map codes. Figure 2 presents a sketch of LLM prompts and an example of numerical codes and see the appendix for a full prompt.

### 4.3 Code-to-Trajectory Decoder

With the interaction-aware codes from the encoder, the code-to-trajectory decoder $\mathcal{D}(\cdot)$ generates vehicle trajectories by aggregating and decoding information between vehicles and interactions. In the decoder, we propose a key design of two-step interaction-aware feature aggregation, which synergizes vehicle interactions with environmental map data, thereby using these interactions to enrich the realism and coherence of vehicle trajectories.

Given the map codes $\mathbf{m}$, the vehicle codes $\mathbf{V}$ and interaction codes $\mathbf{I}$ from the encoder, the overall decoding process can be formulated as

$$\mathbf{E}_{\mathrm{M}}, \mathbf{E}_{\mathrm{V}}, \mathbf{E}_{\mathrm{I}} = \mathcal{F}_{\mathrm{ext}}(\mathbf{m}, \mathbf{V}, \mathbf{I}), \ \widehat{\mathbf{E}}_{\mathrm{V}} = \mathcal{F}_{\mathrm{agg}}(\mathbf{E}_{\mathrm{M}}, \mathbf{E}_{\mathrm{V}}, \mathbf{E}_{\mathrm{I}}), \ \widehat{\mathbb{S}} = \mathcal{F}_{\mathrm{head}}(\widehat{\mathbf{E}}_{\mathrm{V}}), \quad (2)$$

where $\mathcal{F}_{\mathrm{ext}}(\cdot)$ denotes a feature extraction module, $\mathcal{F}_{\mathrm{agg}}(\cdot)$ denotes an attention-based feature aggregation module, $\mathcal{F}_{\mathrm{head}}(\cdot)$ denotes the generation head module for obtaining vehicle attributes and trajectories, and $\mathbf{E}_{\mathrm{M}}, \mathbf{E}_{\mathrm{V}}, \mathbf{E}_{\mathrm{I}}$ are the map lane features, vehicle features and the interaction features, respectively. $\widehat{\mathbf{E}}_{\mathrm{V}}$ denotes the fused features for vehicles and $\widehat{\mathbb{S}}$ is the generated trajectory.

**Feature extraction $\mathcal{F}_{\mathrm{ext}}(\cdot)$.** The feature extraction module $\mathcal{F}_{\mathrm{ext}}(\cdot)$ transforms the numerical codes into initial embeddings for subsequent calculation. For map codes, we retrieve a map $M$ that best fits map code $\mathbf{m}$ from the pre-defined map dataset, which is the same as [7]. The map $M \in \mathbb{R}^{N_L \times N_A}$ consisting of $N_L$ lanes with their $N_A$ attributes, is then passed to the multi-context gating(MCG) blocks [58] obtaining map features $\mathbf{E}_{\mathrm{M}} \in \mathbb{R}^{N_L \times D_L}$ by aggregating neighboring lane information, where $D_L$ is the dimension of each lane feature. For the vehicle codes and interaction codes, we apply MLPs with a position encoding layer for each to obtain their higher-dimensional latent features, that is, $\mathbf{E}_{\mathrm{V}} = \mathrm{MLP}_{\mathrm{V}}(\mathrm{PE}(\mathbf{V})) \in \mathbb{R}^{N \times D_V}$, $\mathbf{E}_{\mathrm{I}} = \mathrm{MLP}_{\mathrm{I}}(\mathrm{PE}(\mathbf{I})) \in \mathbb{R}^{N \times D_I}$, where $\mathrm{PE}(\cdot)$ is the position encoding function, $D_V$ and $D_I$ are the dimensions of extracted vehicle and interaction features.

**Feature aggregation $\mathcal{F}_{\mathrm{agg}}(\cdot)$.** The feature aggregation module aims to fuse the map features and interaction features into vehicle features for subsequent trajectory generation. Based on the intuition that the vehicle interactions are constrained by the road structure and the vehicle states are affected by both the road structure and vehicle interactions, we apply a two-step feature aggregation strategy. First,

we fuse the map feature into the interaction feature and the vehicle feature respectively by multi-head cross-attention operations, that is, $\mathbf{E}'_I = \text{MHATT}_I(\mathbf{E}_I, \mathbf{E}_M, \mathbf{E}_M)$, $\mathbf{E}'_V = \text{MHATT}_V(\mathbf{E}_V, \mathbf{E}_M, \mathbf{E}_M)$, where $\text{MHATT}(q, k, v)$ denotes the multi-head cross-attention functions with query $q$, key $k$, value $v$, $\mathbf{E}'_I$ and $\mathbf{E}'_V$ are the interaction features and the vehicle features after aggregation. Second, we fuse the interaction feature into the vehicle feature to obtain the final vehicle feature, that is, $\widehat{\mathbf{E}}_V = \text{MHATT}_V(\mathbf{E}'_V, \mathbf{E}'_I, \mathbf{E}'_I)$. The final vehicle feature contains both the interaction and map information, which can be manipulated for further trajectory generation.

**Generation head $\mathcal{F}_{\text{head}}(\cdot)$.** The generation head aims to generate vehicle' states and trajectories based on vehicle features. For agent $i$ with agent feature $\widehat{\mathbf{E}}_{V,i}$ and the trajectory type $r_i$ in the vehicle codes, we generate its trajectory positions $\mathbf{S}_i$ through a series of MLP heads. For different trajectory types, we assign different MLP heads. Formally, the generation process is formulated as

$$\widehat{\mathbf{S}}_i = \text{MLP}_{\text{head}, r_i}(\widehat{\mathbf{E}}_{V,i}), \tag{3}$$

where $\text{MLP}_{\text{head}, r_i}$ denotes the assigned $r_i$th heading MLP. We finally assemble all the trajectories $\widehat{\mathbb{S}} = [\widehat{\mathbf{S}}_1, \widehat{\mathbf{S}}_2, \ldots, \widehat{\mathbf{S}}_N]$ together with the map $M$ as output for the scenario generation.

## 4.4 Training

**Generating training samples.** Due to the lack of data directly matching linguistic descriptions with traffic scenarios, we cannot directly optimize the model under ground-truth trajectory supervision using language inputs. As an alternative, we extract map, vehicle, and interaction codes directly from ground-truth trajectories to train the decoder's scenario reduction capability, During the training process, for a ground-truth scenario $S$ derived from real-world datasets, we re-generate the scene by

$$\mathbf{m}, \mathbf{V}, \mathbf{I} = \Psi(\mathbb{S}), \quad \widehat{\mathbb{S}} = \mathcal{D}(\mathbf{m}, \mathbf{V}, \mathbf{I}), \tag{4}$$

where $\Psi(\cdot)$ extracts information from the ground-truth vehicle trajectories to fulfill the codes, including the obtainment and discretization of vehicle speeds, their positions and distances relative to the ego vehicle, and the classification of their trajectory types. The specific computational rules will be mentioned in the appendix. We thus train the decoder $\mathcal{D}(\cdot)$ by minimizing the gap between $S$ and $\widehat{S}$.

**Loss.** We apply a MSE loss $\mathcal{L}_{\text{traj}}(\cdot)$ to minimize differences between generated and ground-truth vehicle trajectories. Furthermore, to enhance the network's sensitivity to trajectory interactions, we additionally supervise the relative distances with the ego vehicle among vehicle trajectories with another MSE loss $\mathcal{L}_{\text{rela}}(\cdot)$. For the $i$th vehicle, the relative distance at last timestep is $\mathbf{d}_i = \mathbf{s}_i^T - \mathbf{s}_1^T$. Formally, the final loss of InteractTraj is presented as

$$\mathcal{L} = \frac{1}{N}\left(\sum_{i=1}^{N} \mathcal{L}_{\text{traj}}(\mathbf{S}_i, \widehat{\mathbf{S}}_i) + \sum_{i=1}^{N} \mathcal{L}_{\text{rela}}(\mathbf{d}_i, \widehat{\mathbf{d}}_i)\right). \tag{5}$$

## 4.5 Discussion

Compared to previous representative traffic trajectory generation method, including CTG [6], CTG++ [16], TrafficGen [4] and LCTGen [7], our method is the first language-conditioned interactive trajectory generation method. (1) At the task level, CTG and CTG++ generate traffic trajectories within the need of vehicles' past trajectory observations. The necessity of collecting past trajectories significantly increases the data generation costs, imposing an extra burden. TrafficGen generates traffic trajectories by only taking a map as input to produce a scenario, resulting in a lack of controllability over the generated trajectories. LCTGen and our methods are specifically designed to generate traffic trajectories based on language conditions, which not only achieves high controllability but also reduces dependency on extensive data sets. (2) Under the same task, compared to LCTGen, our technical novelty comes from two aspects. First and foremost, we propose the interaction codes, corresponding LLM prompts, and interaction-aware feature aggregation which serve as the key to generating interaction-aware traffic trajectories. In contrast, LCTGen does not account for vehicle interactions during trajectory generation. Second, within the vehicle codes, we incorporate a mixed-scale design that both addresses the global type and the detailed movement of vehicle trajectory, which allows the generated trajectories to align with high-level intentions as well as precise positional changes. Conversely, LCTGen only considers the local detailed movement, leading to potential discrepancies between the language descriptions and generation at the high-level trend, such as receiving descriptions to turn left but generating a right turn.

Table 1: Evaluation on trajectory generation realism under WOMD and nuPlan datasets. ↓ indicates lower is better. InteractTraj significantly improves trajectory realism.

| Dataset | Method | mADE ↓ | minADE ↓ | mFDE ↓ | minFDE ↓ | SCR ↓ | HD ↓ |
|---------|--------|--------|----------|--------|----------|-------|------|
| WOMD | TrafficGen | 9.531 | 1.440 | 20.106 | 3.690 | 0.086 | 5.733 |
| | LCTGen | 1.262 | 0.224 | 2.696 | 0.463 | 0.072 | 1.295 |
| | InteractTraj(w/o I) | 1.205 | 0.207 | 2.479 | 0.346 | 0.090 | 1.210 |
| | **InteractTraj** | **1.067** | **0.181** | **2.190** | **0.320** | **0.070** | **1.076** |
| nuPlan | TrafficGen | 9.418 | 1.416 | 19.686 | 3.627 | 0.082 | 5.874 |
| | LCTGen | 1.161 | 0.218 | 2.497 | 0.448 | 0.074 | 1.301 |
| | InteractTraj(w/o I) | 1.108 | 0.181 | 2.277 | 0.323 | 0.070 | 1.150 |
| | **InteractTraj** | **0.962** | **0.160** | **1.987** | **0.321** | **0.067** | **1.129** |

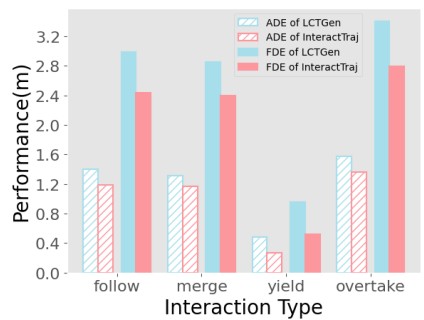

(a) Performances under various interaction types.

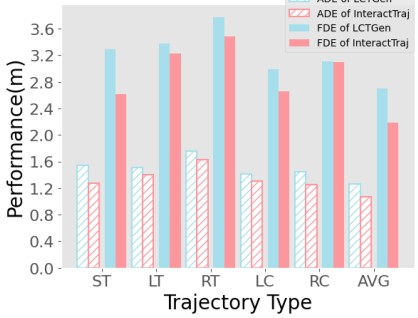

(b) Performances under various trajectory types.

Figure 4: Comparison of model performances under different settings on WOMD. Lower is better. InteractTraj generates more realistic interactive trajectories for different types. ST: straight forward, LT: left turn, RT: right turn, LC: left lane change, RC: right lane change and AVG: average performance.

## 5 Experiments

### 5.1 Dataset and Baseline

We use two datasets, Waymo Open Motion Dataset (WOMD) [59, 60] and nuPlan [61], which both provide real-world vehicle trajectories and corresponding lane maps. We compare our method against two state-of-the-art controllable trajectory generation baselines, TrafficGen [4] and LCTGen [7]. Please refer to the appendix for details on the datasets and the choice of baselines.

### 5.2 Experimental Setup

In the language-to-code encoder, we sample the vehicles' trajectories at 1-second (10 timesteps) intervals to get a $|\mathcal{T}| = 5$ timesteps set. In the code-to-trajectory decoder, the vehicle features $D_V$ and interaction features $D_I$ are set to $256$. During the training process, we train the decoder using the AdamW optimizer [62] with an initial learning rate of $3\mathrm{e}^{-4}$. See more details in the appendix.

### 5.3 Evaluation Metric

Given ground-truth trajectories, we quantify the realism of generated trajectories with 6 metrics: 1) mean average displacement error(mADE); 2) minimum average displacement error(minADE); 3) mean final displacement error(mFDE); 4) minimum final displacement error(minFDE); 5) scenario collision rate(SCR); 6) Hausdorff distance(HD). Detailed formulations of these metrics are provided in the appendix.

### 5.4 Reconstruction-based Evaluation

Since the dataset contains only trajectories and not language-trajectory pairs, we evaluate our methods and baselines quantitatively through a reconstruction approach. For all methods, we generate conditional codes or inputs directly from the ground-truth trajectory instead of the LLM, and then reconstruct the trajectories to assess alignment with the input conditions.

**Quantitative results on all scenarios.** We first evaluate our generated trajectories by comparing them to ground-truth trajectories on the whole dataset. Table 1 compares the performance of InteractTraj with two baseline methods on reconstruction. Since previous methods lack interaction-aware input design, we add one more ablated version of InteractTraj without the interaction code, to have a comparison with the same input information, noted as InteractTraj(w/o **I**). We see that i) our method significantly outperforms previous methods across all the metrics, indicating it generates more realistic

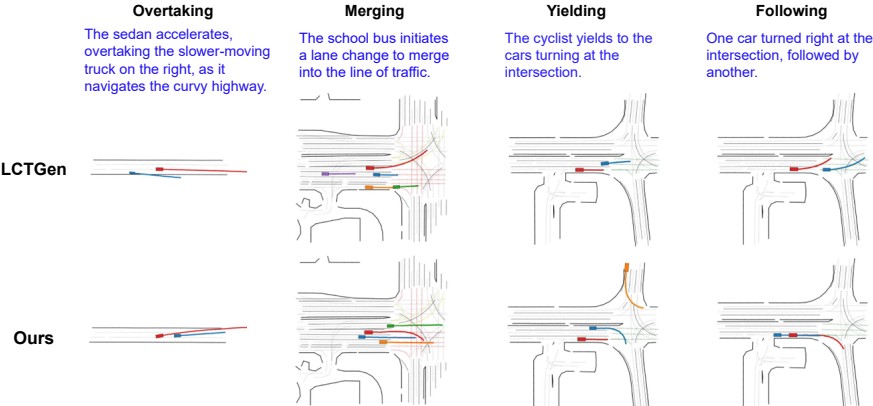

Figure 5: Comparison of model performances under different interaction types. InteractTraj generates trajectories that better align with language descriptions by performing the right vehicle interactions.

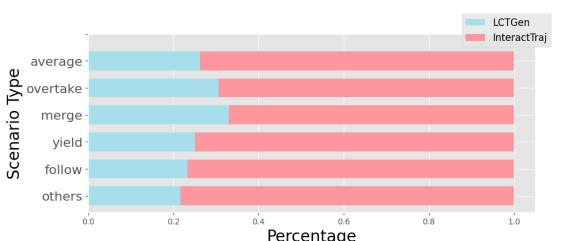

(a) Percentage of users' preference of generated trajectories description of different methods.

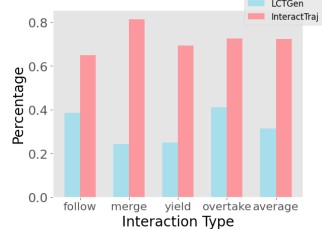

(b) Percentage of users considering the generated scenarios fit the interaction types.

Figure 6: Results of the user study for overall and interaction performances.

scenarios with vehicle interactions. Specifically, our method reduces the mADE/mFDE/HD by 15.4%/18.7%/16.9% compared to SoTA methods; ii) The ablated InteractTraj(w/o **I**) still outperforms previous models, showing the effectiveness of our vehicle code design.

**Quantitative results on scenarios on different interaction types.** To evaluate the performance of model generation on scenarios with trajectory interaction, we test the model on the representative interactive scenarios. The scenarios are mainly categorized into four types according to vehicle interaction: overtaking, converging, yielding and following. Figure 4a shows the method comparison on all types of interactive scenarios with LCTGen. We see that for all types of interactions, InteractTraj significantly reduces the mADE and mFDE, showing a powerful capability to generate realistic interactive trajectories by interaction-aware coding design.

**Quantitative results on scenarios on different trajectory types.** To evaluate the performance of model generation on different individual driving trajectories, we categorize trajectories of the test set into six types according to individual actions: straight, stop, turn left, turn right, left change lanes, left, right change lanes, see detailed rules in the appendix. According to individual action types, we divide the test set and report the average reconstruction performance on every set. Figure 4b shows that InteractTraj's generation results are closer to the ground truth than LCTGen, reflecting we generate trajectories more aligned with language descriptions across different individual driving actions.

It is important to note that the ultimate goal of our method is to generate traffic trajectories from language commands. Previous methods [7, 4] exhibit limitations in that they fail to address the interaction information encoded within languages. Consequently, in the reconstruction-based evaluation, these methods inherently lack interaction information in their inputs. In contrast, our method is capable of incorporating interaction during the generation process, which represents a significant advantage. As a result, our inputs for reconstruction-based evaluation contain more comprehensive information, enabling more realistic and effective trajectory generation.

### 5.5 Language-Conditioned Evaluation

In this section, we compare end-to-end our method with previous methods by analyzing trajectories generated from language descriptions. Given the absence of specific ground-truth trajectories for certain language commands, we employ qualitative evaluation and user studies for assessment.

**Qualitative results.** We compare our model to LCTGen, which also transforms language input into traffic scenarios. We evaluate our methods with that of baseline methods qualitatively given

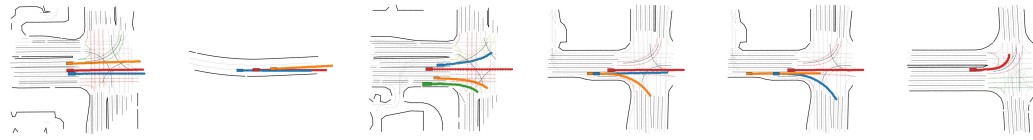

(a) Three cars drive parallel to each other.

(b) Several cars move in platoon formation.

(c) Surrounding vehicles pull over as the ambulance approaches.

(d) Three cars lined up. The car behind makes a right turn at the intersection.

(e) Three cars lined up. The car behind ego car makes a right turn at the intersection.

(f) A car is turning left and moving in a straight line.

Figure 7: Visual analysis for model's performance when dealing with less common interaction types and language ambiguities.

language commands containing different interaction descriptions. Figure 5 provides visualizations of representative user language commands including four types of interaction: vehicle overtaking, vehicle merging, vehicle yielding, and vehicle following. We see that compared to previous work, the scenarios generated by InteractTraj better align with language descriptions with the help of interaction-aware code representation, while the previous method can not perform the corresponding interactive action since it generates trajectories of each agent independently.

**Generalisation capability analysis.** Our model demonstrates the ability to generate compliant scenarios even when handling less common or emergent interaction types as shown in the left part of Figure 7. We test interaction scenarios not mentioned explicitly in the prompt, including uncommon cases of parallel driving (Figure 7a), platooning (Figure 7b), and pulling over (Figure 7c). The results show that our method can effectively interpret and translate these less common interaction types, producing scenarios that align with expected behaviors without retraining. This is primarily due to the robust generalization capacity of the LLM used in our encoding process. The decoding process is also equipped with strong generalization abilities due to extensive training with massive numerical codes, which would translate these codes into trajectories and generate compliant scenarios.

In addressing the potential ambiguities in language descriptions, our approach leverages the reasoning capabilities of LLM without introducing additional prioritized proximity. Ambiguities typically arise in two main forms: unclear references to objects and contradictions within the language instructions.

1) For cases where the object reference is ambiguous, the LLM interprets the linguistic input and converts it into numerical codes that correspond to one of the plausible meanings. As illustrated in Figure 7d, the LLM assigns the label "the car behind" to a randomly selected vehicle. However, when the description is more specific, such as "the car behind ego car" (Figure 7e), the LLM accurately resolves the reference and appropriately handles the description.

2) For cases where there are self-contradictory language requirements, the LLM generates scenarios that partially align with the instructions. For instance, as shown in Figure 7f, the instruction presents an inherent contradiction. The LLM resolves this by prioritizing one part of the instruction while disregarding the other. To better tackle language ambiguity, a potential solution involves introducing LLM-human interaction to iteratively verify language descriptions.

**Controllability analysis.** In Figure 8, it is evident that each generated scenario adheres to the provided linguistic descriptions when there are variations in the details. This highlights the strong controllability of InteractTraj in accurately manipulating vehicle behaviors, ensuring that the generated scenarios align with user-specified control commands. The precision is achieved through the LLM's robustness in understanding and applying detailed instructions based on different linguistic inputs.

**User study.** We conduct two user studies on WOMD to qualitatively assess the language-conditioned traffic scenario generation capabilities of InteractTraj from two perspectives: 1) overall generation performance and 2) vehicle interaction performance. GPT-4 is used to generate descriptions of different interaction types as input language descriptions, see appendix for the details.

In the first user study, each user is given forty language commands and corresponding trajectories generated by models and is asked to choose the one trajectory that better fits the language description. Figure 6a shows the results of users' preferences for scenarios generated by LCTGen and InteractTraj. We see that i) for each interaction type, significantly more users prefer scenarios generated by InteractTraj over those produced by LCTGen; ii) on average, 73.7% responses are more favorable to the scenarios generated by InteractTraj, and our model achieves at least 66.9% support on all

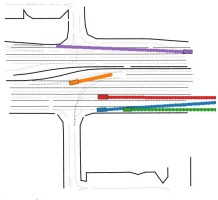
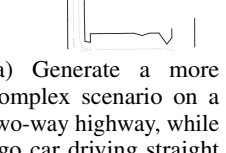
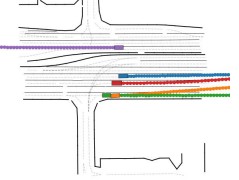
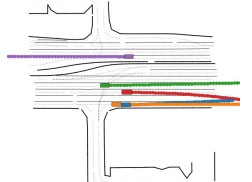
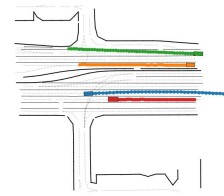

| (a) Generate a more complex scenario on a two-way highway, while ego car driving straight forward. | (b) Generate a more complex scenario on a two-way highway, while ego car making a left-lane change.. | (c) Generate a more complex scenario on a two-way highway, while ego car is making a right-lane change. | (d) Generate a more complex scenario on a two-way highway, while ego car being overtaken. |
|---|---|---|---|

Figure 8: Visualization results for model's controllability

Table 2: Ablation study on proposed code and network designs evaluated on WOMD. All designs are beneficial.

| Model | IC | RD | RP | $\mathcal{L}_{\mathrm{rd}}$ | mADE ↓ | mFDE ↓ | SCR ↓ |
|---|---|---|---|---|---|---|---|
| (a) |  | ✓ | ✓ | ✓ | 1.205 | 2.479 | 0.346 |
| (b) | ✓ |  | ✓ | ✓ | 1.167 | 2.442 | 0.084 |
| (c) | ✓ | ✓ |  | ✓ | 1.194 | 2.475 | 0.080 |
| (d) | ✓ | ✓ | ✓ |  | 1.165 | 2.446 | 0.080 |
| **Ours** | ✓ | ✓ | ✓ | ✓ | **1.067** | **2.190** | **0.070** |

Table 3: Ablation study on the granularity of the discretization of the interaction codes on WOMD.

| Model | Gap | Areas | mADE ↓ | mFDE ↓ | SCR ↓ |
|---|---|---|---|---|---|
| (e) | 10 | 6 | 1.087 | 2.228 | 0.074 |
| (f) | 20 | 6 | 1.117 | 2.299 | 0.070 |
| (g) | 15 | 4 | 1.237 | 2.810 | 0.071 |
| (h) | 15 | 8 | 1.069 | 2.190 | 0.071 |
| **Ours** | 15 | 6 | **1.067** | **2.190** | **0.070** |

sub-categories. This reflects that InteractTraj has a stronger capability at generating interactive trajectories than LCTGen, and excels in representing interaction aspects of language descriptions.

The second user study contains fifty questions covering different interaction types, and users are asked to answer whether the scenarios generated fulfill the corresponding textual descriptions. The results are shown in Figure 6b. We see that i) for each interaction type, significantly more users consider the scenarios generated by InteractTraj to fulfill the requirements given by language descriptions; ii) on average, $72.4\%$ positive responses consider that InteractTraj generates scenarios with required interactions, while LCTGen only have $31.5\%$ positive responses in average. This reflects that InteractTraj effectively extracts the interaction information in the descriptions, and generates sufficiently satisfying traffic scenarios.

### 5.6 Ablation Study

**Effect of proposed code and network design.** We conduct the ablation study based on the reconstruction evaluation to evaluate the effectiveness of proposed designs, including a) the addition of whole interaction codes (IC); b) the relative distance in interaction codes (RD); c) the relative position in interaction codes (RP); d) the relative distance loss $\mathcal{L}_{\mathrm{rd}}$. Table 2 presents the results. We see that all designs are beneficial to a more realistic trajectory generation.

**Effect of the setting of hyper-parameters.** We conduct an ablation study of the granularity of the discretization of the relative distances and relative positions, specifically including e), f) the interval gap used for discretizing relative distances; g), h) the number of areas used for discretizing relative distances, as shown in Table 3. We see that our current parameter choices achieve the best results.

## 6 Conclusion

We propose InteractTraj, a novel interaction-aware language-guided traffic scenario generation model. Our core idea is to convert language descriptions into multi-level codes and generate trajectories by attention-based information aggregation. Experiments show that InteractTraj effectively reproduces real-life scenario distribution and generates scenarios aligned with language description.

**Limitations and future work.** This work focuses on generating trajectories of only vehicles and the generation of maps is limited by the map library. In the future, we plan to extend the work to more types of traffic participants and more flexible map generation. We also plan to apply the generated traffic scenarios to the training of autonomous driving systems by expanding the motion dataset.

## 7 Acknowledgment

This research is supported by NSFC under Grant 62171276 and the Science and Technology Commission of Shanghai Municipal under Grant 21511100900, 22511106101, and 22DZ2229005. Special thanks to Lightwheel AI for the support.

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

# Appendix

In the appendix, we further explain the experiment settings, the definition of metrics used for evaluation, complete prompts used to transform language input into codes, the design of the trajectory analyzing module, and some examples and illustrations extracted from our user studies.

## A    Additional Experiment Settings

**Dataset.** We use WOMD and nuPlan as the datasets for our experiments. For WOMD, we adopt $68,000$ scenarios for training and $2500$ scenarios for testing, and for nuPlan, we selected $82,122$ scenarios for training and $20,756$ scenarios for testing from the whole dataset. Following the setting in previous works [7], for each scenario, we keep a maximum number of 384 lanes and 32 vehicles. We generate a 5-second trajectory at 10 fps, that is, T=50 timesteps.

**Baseline.** We consider two existing controllable trajectory generation baselines, TrafficGen [4] and LCTGen [7]. TrafficGen is the first approach that generates vehicle trajectories purely from empty maps without relying on vehicle past states. LCTGen is the latest open-sourced language-guided vehicle trajectory generation approach. Both models represent state-of-the-art performance in trajectory generation. We make some adjustments to the baseline models to standardize the number of vehicles compared in the scene and the length of the predicted trajectories, ensuring a fair comparison.

**Experimental setup.** In the language-to-code encoder, we discretize inter-vehicle distances at 15-meter intervals and discretize speeds at 2.5-meter-per-second intervals. We sample the vehicles' trajectories at 1-second (10 timesteps) intervals to get a $|\mathcal{T}| = 5$ timesteps. We use MCG blocks with 5 layers for lane feature extraction, a 2-layer transformer with 4 heads is used for decoding vehicle and interaction queries, and a cross-attention module with 8 heads is used for fusing interaction features into vehicle features. An MLP with a latent dimension of 512 is finally used for trajectory generation. Only the code-to-trajectory decoder needs to be trained as the LLM in our model encoder does not need training. It takes about 12 hours for 100 epochs on 4 NVIDIA GeForce RTX040 GPUs for the decoder training process. During the inference phase, it takes about 30 seconds to generate a scenario, which includes about mainly 30 seconds to receive the response from the LLM and about 0.2 seconds to generate corresponding trajectories.

## B    Evaluation Metric Details

For each trajectory prediction and the corresponding ground truth trajectory, we measure their similarity using the following six metrics. Mean average displacement error(mADE), minimum average displacement error(minADE), mean final displacement error, minimum final displacement error(minFDE), and Hausdorff distance(HD) measure the gap between the projected paths and the actual paths, while the collision rate reflects the rationalization of the generated scenarios. For all metrics, smaller values mean that the generated trajectories are closer to the ground truth values, and therefore the corresponding model can be considered to generate scenarios closer to the real-life data distribution.

**Symbol Definition**

- Let $N$ be the number of agents in the scenario.

- Let $\mathbb{G}$ be the set of scenarios used for testing, let $N_G = \|\mathbb{G}\|$ be the number of scenarios tested.

- Let $T$ be the overall timesteps.

- Let $\mathbb{S}_g = [\mathbf{S}_{g1}, \mathbf{S}_{g2}, \ldots, \mathbf{S}_{gN}] \in \mathbb{R}^{N \times T \times 2}$ be the set of ground-truth trajectories contained in a scenario $g \in \mathbb{G}$, where $\mathbf{S}_{gi} = [s_{gi}^1, s_{gi}^2, \cdots, s_{gi}^T] \in \mathbb{R}^{T \times 2}, \forall i \in \{1, \ldots, N\}$ denotes the ground-truth trajectory of agent $i$ over $T$th timesteps.

- Let $\widehat{\mathbb{S}_g} = [\widehat{\mathbf{S}_{g1}}, \widehat{\mathbf{S}_{g2}}, \ldots, \widehat{\mathbf{S}_{gN}}] \in \mathbb{R}^{N \times T \times 2}$ be the set of predicted trajectories contained in the scenario $g \in \mathbb{G}$, where $\widehat{\mathbf{S}_{gi}} = [\widehat{s_{gi}^1}, \widehat{s_{gi}^2}, \cdots, \widehat{s_{gi}^T}] \in \mathbb{R}^{T \times 2}, \forall i \in \{1, \ldots, N\}$ denotes the predicted trajectory of agent $i$ over $T$ timesteps.

**Mean Average Displacement Error(mADE):** mADE refers to the average mean square error (MSE) between all predicted points and the ground-truth points over all the trajectories in a scenario. Its average value over the entire test set is defined as:

$$\text{mADE}(\mathbb{G}) = \frac{1}{N_G} \sum_{g=1}^{N_G} \left( \frac{1}{N} \sum_{i=1}^{N} \left( \sum_{t=1}^{T} \|s_{gi}^t - \widehat{s_{gi}^t}\|^2 \right) \right)$$

**Mean Final Displacement Error(mFDE):** mFDE refers to the average distance between the predicted final destination and the ground-true final destination over all the trajectories in a scenario. Its average value over the entire test set is defined as:

$$\text{mFDE}(\mathbb{G}) = \frac{1}{N_G} \sum_{g=1}^{N_G} \left( \frac{1}{N} \sum_{i=1}^{N} \|s_{gi}^T - \widehat{s_{gi}^T}\|^2 \right)$$

**Minimum Average Displacement Error(minADE):** minADE refers to the minimum mean square error (MSE) between all predicted and ground-truth trajectories in a scenario. Its average value over the entire test set is defined as:

$$\text{minADE}(\mathbb{G}) = \frac{1}{N_G} \sum_{g=1}^{N_G} \left( \min_{i \in N} \left( \sum_{t=1}^{T} \|s_{gi}^t - \widehat{s_{gi}^t}\|^2 \right) \right)$$

**Minimum Final Displacement Error(minFDE):** minFDE refers to the minimum distance between the predicted final destination and the ground-true final destination over all the trajectories in a scenario. Its average value over the entire test set is defined as:

$$\text{minFDE}(\mathbb{G}) = \frac{1}{N_G} \sum_{g=1}^{N_G} \left( \min_{i \in N} \|s_{gi}^T - \widehat{s_{gi}^T}\|^2 \right)$$

**Hausdorff distance(HD):** HD refers to the Hausdorff distance between all predicted and ground-truth trajectories, which measures how far two trajectories of a metric space are from each other when viewed as point sets in a scenario. Its average value over the entire test set is defined as:

$$\text{HD}(\mathbb{G}) = \frac{1}{N_G} \sum_{g=1}^{N_G} \left( \frac{1}{N} \sum_{i=1}^{N} d_H(S_{gi}, \widehat{S_{gi}}) \right),$$

where the Hausdorff distance is defined as

$$d_H(X, Y) = \max\{sup_{x \in X} \text{MSE}(x, Y), sup_{y \in Y} \text{MSE}(X, y)\}$$

**Scenario collision rate(SCR):** Among the predicted trajectories, We note the bounding box of agent $i$ as $\text{box}_i^t$ at time $t$. For a pre-defined threshold $\delta$, agent $i$ and agent $j$ are considered to be collided if $\forall t \in T, \text{IoU}\left(\text{box}_i^t, \text{box}_j^t\right) > \delta$, where $\text{IoU}(\cdot, \cdot)$ denotes the intersection over union between two objects. We thus define scenario collision rate as

$$\text{SCR}(\mathbb{G}) = \frac{1}{N_G} \sum_{g=1}^{N_G} \left( \frac{2}{N \times (N-1)} \sum_{i=1}^{N} \sum_{j>i}^{N} \mathbb{1}(\forall t \in T, \text{IoU}\left(\text{box}_i^t, \text{box}_j^t\right) > \delta) \right)$$

## C   LLM Prompt Details

Here we show our prompt used to generate vehicle codes, map codes, and interaction codes.

> You are a faithful format converter that translates natural language descriptions to a fixed-form format to appropriately describe the scenario with motion action. You also need to output an appropriate map description that supports this scenario. Your ultimate goal is to generate

realistic traffic scenarios that faithfully represent natural language descriptions and normal scenes that follow the traffic rules.

Answer with a list of codes describing the attributes of each of the vehicles and the interactions within the events in the scenario.

Desired format:

Summary: summarize the scenario in short sentences, including the number of vehicles. Also, explain the underlying map description.

Explanation: If there are interaction behaviors concluded in the requirements, first explain the meaning of these terms such as the behaviors of the agent involved. Then explain for each group of vehicles why they are put into the scenario and how they fulfill the requirement in the description.

Vehicle Code: A list of codes of length ten, describing the attributes of each of the vehicles in the scenario, only output the values without any text:
- 'V1': [,,,,,,,,,]
- 'V2': [,,,,,,,,,]
- 'V3': [,,,,,,,,,]

Map Code: A code of length six describing the map attributes, only output the values without any text:
- 'Map': [,,,,,]

Interaction Code: For each agent, generate two codes, each of length five. The first code represents the relative distance of the vehicle with respect to the ego car, and the second one represents the relative position of the ego car. Only output the values without any text:
- 'I1': [,,,,] | [,,,,]
- 'I2': [,,,,] | [,,,,]

Meaning of the vehicle code attribute:
- dim 0: 'pos': [-1,5] - The relative position of the vehicle with respect to ego car in the order of [0 - 'front', 1 - 'front right', 2- 'back right', 3 - 'back', 4 - 'back left', 5 - 'front left']. -1 if the vehicle is the ego vehicle.
- dim 1: 'distance': [0,3] - the distance range index of the vehicle towards the ego vehicle; the range is from 0 to 60 meters with 15 meters intervals. 0 if the vehicle is the ego vehicle. For example, if the distance value is less than 15 meters, then the distance range index is 0.
- dim 2: 'direction': $[0, 3]$ - the direction of the vehicle relative to the ego vehicle, in the order of 0-'parallel same', 1-'parallel opposite', 2-'perpendicular up', 3-'perpendicular down'. 0 if the vehicle is the ego vehicle.
- dim 3-8: 'speed trend': [0,8] - speed of the agent in future 5 seconds(including initial speed and final speed) with consecutive dimensions having a time interval of 1s. Velocity is categorized into nine grades from 0-8, with smaller grades resulting in higher speeds. The range is from 0 to 20 m/s with a 2.5 m/s interval. For example, 20m/s is in the range of 8, therefore the speed value is 8.
- dim 9: 'action': [0,5] - category of vehicle behavior during this time. The vehicle's action is divided into six types: [0 - 'stop', 1 - 'straight', 2 - 'left-turn', 3 - 'right-turn', 4 - 'left-change-lane', 5 - 'right-change-lane']

Meaning of the Map code attributes:
- dim 0-1: 'parallel lane count': 2-dim. The first dim is the number of parallel same-direction lanes of the ego lane, and the second dim is the number of parallel opposite-direction lanes of the ego lane.
- dim 2-3: 'perpendicular lane count': 2-dim. The first dim is the number of perpendicular upstream-direction lanes, and the second dim is the number of perpendicular downstream-direction lanes.
- dim 4: 'dist to intersection': 1-dim. the distance range index of the ego vehicle to the intersection center in the x direction, range is from 0 to 60 meters with 15 meters intervals. -1 if there is no intersection in the scenario.
- dim 5: 'lane id': 1-dim. the lane ID of the ego vehicle, counting from the rightmost lane of the same-direction lanes, starting from 1. For example, if the ego vehicle is in the rightmost lane, then the lane id is 1; if the ego vehicle is in the leftmost lane, then the lane id is the number of the same-direction lanes.

Meaning of the interaction code attributes:

- Each code has a length of 5, representing the relative distance and direction of the vehicle with ego car respectively. Two neighboring values in each code have an interval of one second, so it can be used to represent the tendency of the trajectory relative to the ego car.
- The values of the first code represents the distance of this car relative to the ego car. It is divided into five bins from 0-5, with each bin has an interval of five meters. When the value is 0, it means that the car is very close to the ego car. The farther away this vehicle is from the ego car, the larger the value is. When the distance is more than 25 meters, the value is always set to 5.
- The values of the second code pair indicates the distance of the car relative to the ego car. For each dimension, 0 means that the vehicle is in front of the ego car directly, 1 represents the right front, 2 represents the right rear, 3 represents the rear, 4 represents the left brear and 5 represents the left front.

Traffic rules that you should obey when creating representation for traffic scenes:
- When the car drives to an intersection, it should slow down whether it is turning or going straight ahead.
- If another vehicle is close to the ego vehicle and passes in front of the ego vehicle, for example, driving from the left front to the right front, the vehicle should stop and wait for the other car's passing by.
- When changing lanes, pay attention to whether there are other vehicles on the near left or near right side of the car, and if there are, keep driving in the current direction.
- When the vehicle turns left, it should pay attention to the left rear and make sure that there are no other vehicles, keep straight ahead otherwise.
- When the vehicle turns right, it should pay attention to the left rear and make sure that there are no other vehicles, keep straight ahead otherwise.
- The car should not change lanes to the left when it is in the far left lane.
- The car should not change lanes to the right when it is in the far right lane.

Some nomenclature so you can better understand how vehicles interact with each other to represent their trajectories and movement trends:
- overtake: When another vehicle directly in front of the vehicle is moving too slowly, ego vehicle can overtake the vehicle in front of it by changing lanes to the left or right and accelerating. Generally, that car is directly in front of the ego car at the beginning and drives slower than the ego car. In the end, it will be on the rear side of the ego car.
- merge: There are more than two cars on the road. One vehicle is first selected as the ego car and is kept in a straight line and keeps its speed throughout the process. The other cars are in the lane adjacent to the ego car, keeping a relatively close distance at first. These cars in turn merge into the ego car's lane by changing lanes. For example, the vehicle to the left of the ego car changes lanes to the right, and the vehicle to the right of the ego car changes lanes to the left. If the car wants to change lanes to the left and there is another vehicle in the left lane, it should decelerate and wait for that vehicle to pass before making the left lane change. The same is true when a vehicle is changing lanes to the right. During the process, you should keep a safe distance with other cars in order not to collide.
- rear-ending/rear-end collision: When the first car (for example, to avoid someone crossing the street) makes a sudden deceleration, and the car behind collides with it. Generally, the second car will be behind the ego car at the very beginning, some distance away and faster than the ego car. But at the end, that car's position will almost overlap with the ego car and come to a stop.

Transform the query sentence to the vehicle codes strictly following the rules below:
- Ensure the code of each vehicle has a length of 10.
- Focus on the interactions between the vehicles in the scenario.
- Focus on realistic action generation of the motion to reconstruct the query scenario.
- First determine the action type of the agent to fill index 9.
- Pay particular attention to the type of trajectories generated, and the corresponding trajectories are generated according to the inferred trajectory class.
- Follow traffic rules to form a fundamental principle in most road traffic systems to ensure the safety and smooth operation of traffic. You should incorporate this rule into the behavior of our virtual agents (vehicles).

- For speed and distance, convert the unit to m/s and meter, and then find the interval index in the given range.
- Describe the initialization status of the scenario.
- During generation, the number of vehicles is within the range of [1, 32].
- The maximum distance should not exceed 60m (index 1).
- The maximum speed should not exceed 20m/s (index 3-8).
- Always generate the ego vehicle first (V1).
- Always assume the ego car is in the center of the scene and is driving in the positive x direction.
- In the input descriptions, regard V1, Vehicle 1 as the ego vehicle. All the other vehicles are the surrounding vehicles. For example, for "Vehicle 1 was traveling southbound", the ego car is Vehicle 1.
- If the vehicle is stopping, its speed should be 0m/s (index 3-8). Also, if the first action is 'stop', then the speed should be 0m/s (index 3-8).
- If vehicle move in slow speed, the speed should be less than 2.5m/s (index 1) or 5m/s (index 2).
- Try to increase the variation of the placement and motion of the vehicles under the constraints of the description.
- Both turning and lane-changing processes need to reflect changes in speed. For example, during turning and lane changing the vehicle needs to undergo a deceleration process to ensure safety, and after the maneuver is completed the vehicle will reaccelerate. The change in speed is reflected in the 'speed trend' dimensions in the vehicle code.
Generate the map code following the rules below:
- If there is vehicle turning left or right, there must be an intersection ahead.
- If the car was going to change lanes to the left, he couldn't have been in the far left lane. If the car was going to change lanes to the right, he couldn't have been in the far right lane.
- Should at least have one lane with the same direction as the ego lane; i.e., the first dim of Map should be at least 1. For example, if this is a one-way two-lane road, then the first dim of the Map should be 2.
- Regard the lane at the center of the scene as the ego lane.
- Consider the ego car's direction as the positive x direction. For example, for "V1 was traveling northbound in lane five of a five-lane controlled access roadway", there should be 5 lanes in the same direction as the ego lane.
- The generated map should strictly follow the map descriptions in the query text. For example, for "Vehicle 1 was traveling southbound", the ego car should be in the southbound lane.
- If there is an intersection, there should be at least one lane in either the upstream or downstream direction.
- If there is no intersection, the distance to the intersection should be -1.
- There should be a vehicle driving vertically to the ego vehicle in the scene only when there is an intersection in the scene. For example, when the road is just two-way, there should not be any vehicle driving vertically to the ego vehicle.
- If no intersection is mentioned, generate intersection scenario randomly with real-world statistics.
Generate the interaction codes following the rules below:
- Ensure the interaction code has a length of 10.
- Generate an interaction code for each of the vehicles, with the first code remaining all zero since the ego car always overlaps itself.
- Interpreting the relative position of a vehicle as a trajectory over a future period of time.
- Determine their mutual speed and direction from the relative positions of the cars.
- If the relative position crosses the vehicle, such as driving from the left rear to the right front of the vehicle, or from the right rear to the left front, there is a possibility of a collision, and vice versa.
- If another vehicle drives from the left front of the ego vehicle all the way to the right front, or the right front to the left front, it is possible for the ego vehicle to remain stopped waiting for the other vehicle to go first.

> - Reasoning about the relative distance and position of two cars based on the information of the map. For example, if there is only one lane, there are only other cars that may be in front of and behind the car, and there cannot be any other cars on the left or right side.

## D  Trajectory Analyzing Module

During the training process, for a ground-truth scenario $S$ derived from real-world datasets, we re-generate the scene by

$$\mathbf{m}, \mathbf{V}, \mathbf{I} = \Psi(\mathbb{S}), \quad \widehat{\mathbb{S}} = \mathcal{D}(\mathbf{m}, \mathbf{V}, \mathbf{I}), \tag{6}$$

where $\Psi(\cdot)$ denotes an analyzing module that discretizes ground-truth trajectories into vehicle, map and interaction codes. The length of each code and the meaning of each dimension are the same as the codes generated by the language-to-code module, as described in the previous sections and the detailed prompts.

**Map code**: $\Psi(\cdot)$ calculates the number of lanes included in the map both vertically and horizontally, the distance of the ego car from the intersection, and the lane on which the ego car is located on to fill up the map code.

**Interaction code**: For each vehicle other than the ego car, $\Psi(\cdot)$ samples its ground-truth trajectory at regular intervals, using these sampled time points to compute the relative position and relative distance to represent its trend of movement relative to the ego vehicle. In practice, we sample the trajectory points every second and remove outliers concluded. For each sampled point, $\Psi(\cdot)$ discretizes the distance and position to fill the corresponding part of the interaction code. The relative distance is divided by a fixed interval as the relative distance code; for the relative position, we use the ego car as the origin and the heading of the ego car as the positive direction of the x-axis, and the whole space is divided into several equal-partitioned regions. The relative position code can be thus represented by the serial number of the grid in which the vehicle is located. For example, when we select the number of areas to be 6, then the space is divided into six equal-partitioned parts, with code 0 representing the area directly in front, code 1 representing the front-right area, code 2 representing the front-back area, code 3 representing the rear area, code 4 representing the back-left area, and code 5 representing the front-left area. The length of the interval and the number of areas chosen are discussed in the ablation study.

**Vehicle code**: The computation of relative position and the discretization of speed are similar to the previous parts, and we also sample at fixed intervals to represent the trend of the vehicle's speed change. In addition, we categorize vehicles into six types based on their trajectory: stop, straight ahead, left turn, right turn, left lane change, and right lane change. Likewise, we note $\mathbf{S}_i = [s_i^1, s_i^2, \cdots, s_i^T] \in \mathbb{R}^{T \times 2}, \forall i \in \{1, \ldots, N\}$ be the ground-truth trajectory of agent $i$ over $T$ timesteps, $\mathbf{V}_i = [v_i^1, v_i^2, \cdots, v_i^T] \in \mathbb{R}^T, \forall i \in \{1, \ldots, N\}$ be the ground-truth velocity of agent $i$ over $T$ timesteps and $\mathbf{H}_i = [h_i^1, h_i^2, \cdots, h_i^T] \in \mathbb{R}^{T \times 2}, \forall i \in \{1, \ldots, N\}$ be the ground-truth heading of agent $i$ over $T$ timesteps.

We categorize the trajectory types according to the following rules.

**Stop:** Given pre-defined distance threshold $\delta_d$ and velocity threshold $\delta_v$, Agent $i$ is considered to be stopping if $\forall (t, t') \in T^2, \|s_i^t - s_i^{t'}\| \leq \delta_d$ and $\forall (t, t') \in T^2, \|v_i^t - v_i^{t'}\| \leq \delta_v$. In practice, $\delta_d$ is set to be 1 meter, and $\delta_s$ is set to be 0.2 meters per second.

**Left turn:** Given pre-defined lane-width $l_w$ and angle threshold $\delta_a > 0$, agent $i$ is considered to be making a left turn if it's heading sharply towards its left and its displacement in the direction perpendicular to its heading is greater than the lane-width: $\left\| (s_i^T - s_i^0) - \left( (s_i^T - s_i^0) \cdot h_i^0 \right) h_i^0 \right\| \geq l_w$, $\forall (t, t') \in T^2, \left( h_i^t \times h_i^{t'} \right)_z \geq 2 \times \delta_a$. In practice, $\delta_a$ is set to be $\frac{\pi}{12}$.

**Right turn:** Given pre-defined lane-width $l_w$ and angle threshold $\delta_a > 0$, agent $i$ is considered to be making a right turn if it's heading sharply towards its right and its displacement in the direction perpendicular to its heading is greater than the lane-width: $\left\| (s_i^T - s_i^0) - \left( (s_i^T - s_i^0) \dot{h}_i^0 \right) h_i^0 \right\| \geq l_w$, $\forall (t, t') \in T^2, \left( h_i^t \times h_i^{t'} \right)_z \geq 2 \times \delta_a$. In practice, $\delta_a$ is set to be $\frac{\pi}{12}$.

**Left lane change:** Given pre-defined lane-width $l_w$ and angle threshold $\delta_a > 0$, agent $i$ is considered to be making a left lane change if it's heading slightly towards its left and its displacement in the direction perpendicular to its heading is greater than a certain multiple of lane-width: $\left\| (s_i^T - s_i^0) - \left( (s_i^T - s_i^0) \dot{h}_i^0 \right) h_i^0 \right\| \geq l_w, \forall (t, t') \in T^2, \left( h_i^t \times h_i^{t'} \right)_z \in [\delta_a, 2 \times \delta_a]$. In practice, $\delta_a$ is set to be $\frac{\pi}{12}$.

**Right lane changeL** Given pre-defined lane-width $l_w$ and angle threshold $\delta_a > 0$, agent $i$ is considered to be making a right lane change if it's heading slightly towards its right and its displacement in the direction perpendicular to its heading is within a certain multiple of lane-width: $\left\| (s_i^T - s_i^0) - \left( (s_i^T - s_i^0) \dot{h}_i^0 \right) h_i^0 \right\| \geq l_W, \forall (t, t') \in T^2, \left( h_i^t \times h_i^{t'} \right)_z \in [-2 \times \delta_a, -\delta_a]$. In practice, $\delta_a$ is set to be $\frac{\pi}{12}$.

**Straight** Most of the remaining agent trajectories remain within the lane width, and the variation in the direction of the agent is within a proper range, and thus these agents are considered to be driving straight forward. There remains a small percentage of agents that have missing values or outliers in their trajectories, which are given a default trajectory type and are masked out in the following processes.

# E   Additional User Study Details

We use GPT-4 to generate a series of natural language descriptions of traffic scenes containing vehicles, e.g., "The car speeds up to pass the vehicle ahead of it.", on which later InteractTraj and LCTGen are used to generate scenarios respectively. Users are invited to evaluate the generated scenarios depend on different requirements. We balance the frequency of each type of interaction event in these descriptions as much as possible.

The first user study contains forty language commands, and for each command, the two models generate the corresponding trajectory. Each user is asked to choose the trajectory that better fits the language description. A total of 31 interviewees participated in the research, with a total of 1240 samples. In the second user study, we have fifty language descriptions that cover and emphasize the most representative interaction types. For each description, users are asked to answer whether the scenarios generated fulfill the corresponding textual descriptions from their perspectives. The answer can be simultaneously positive or negative for either of the questions. A total of 28 users participated in the study, with a total of 1400 samples.

In this section, we give two examples for each user study as an illustration.

**User study 1: overall generation performance** The first user study contains forty language commands, and for each command, LCTGen and InteractTraj are used to generate corresponding trajectories respectively. Each user is asked to choose from either of them that better fits the language description. Figures 9 and 10 illustrate two language descriptions and the corresponding scenarios generated.

**User study 2: vehicle interaction performance** The second user study contains fifty language commands and each command represents a representative type of interaction previously mentioned. For each command, LCTGen and InteractTraj are used to generate corresponding trajectories respectively and each user is asked to evaluate whether the scenarios fit the interaction behaviors given by the linguistic description. 11 and 12 illustrate two language descriptions and the corresponding scenarios generated.

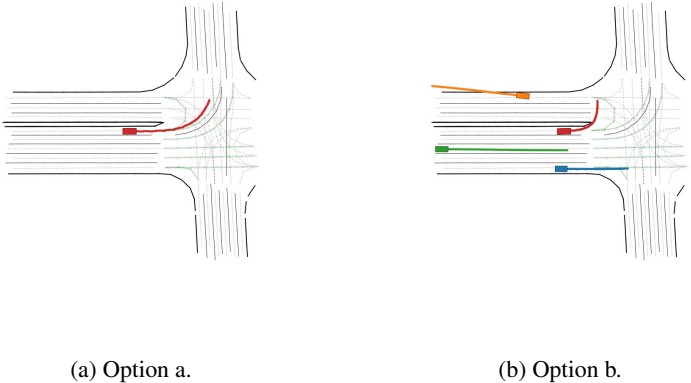

(a) Option a.        (b) Option b.

Figure 9: Description: The vehicle slows down and turns left at the intersection.

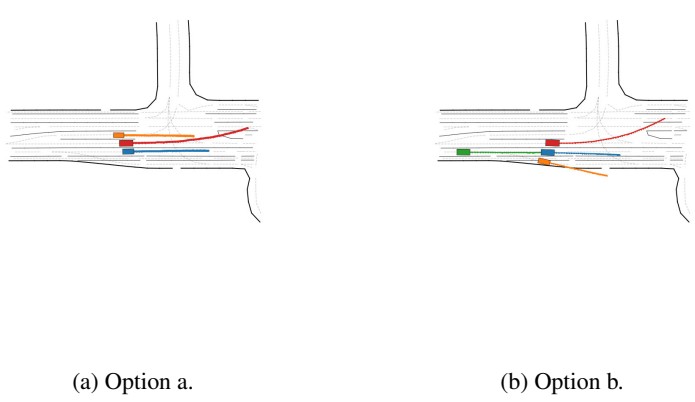

(a) Option a.        (b) Option b.

Figure 10: Description: The lead vehicle signals a lane change, prompting the following cars to adjust their speeds and positions accordingly


Figure 11: Yielding: The sedan yields to the oncoming ambulance.

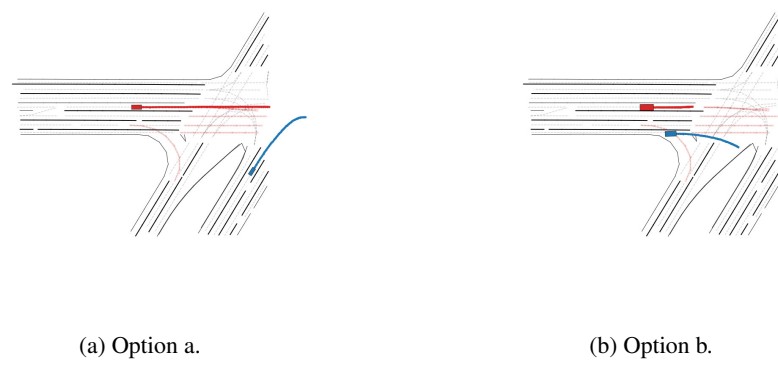

(a) Option a.          (b) Option b.

Figure 12: Merging: As the car approached the intersection, it slowed down, allowing the motorcycle to merge into the space.

2. **Limitations**

   Question: Does the paper discuss the limitations of the work performed by the authors?

   Answer: [Yes]

   Justification: We have included a discussion of the current limitations and directions for future work in section 6.

   Guidelines:

   - The answer NA means that the paper has no limitation while the answer No means that the paper has limitations, but those are not discussed in the paper.
   - The authors are encouraged to create a separate "Limitations" section in their paper.
   - The paper should point out any strong assumptions and how robust the results are to violations of these assumptions (e.g., independence assumptions, noiseless settings, model well-specification, asymptotic approximations only holding locally). The authors should reflect on how these assumptions might be violated in practice and what the implications would be.
   - The authors should reflect on the scope of the claims made, e.g., if the approach was only tested on a few datasets or with a few runs. In general, empirical results often depend on implicit assumptions, which should be articulated.

- The authors should reflect on the factors that influence the performance of the approach. For example, a facial recognition algorithm may perform poorly when image resolution is low or images are taken in low lighting. Or a speech-to-text system might not be used reliably to provide closed captions for online lectures because it fails to handle technical jargon.
- The authors should discuss the computational efficiency of the proposed algorithms and how they scale with dataset size.
- If applicable, the authors should discuss possible limitations of their approach to address problems of privacy and fairness.
- While the authors might fear that complete honesty about limitations might be used by reviewers as grounds for rejection, a worse outcome might be that reviewers discover limitations that aren't acknowledged in the paper. The authors should use their best judgment and recognize that individual actions in favor of transparency play an important role in developing norms that preserve the integrity of the community. Reviewers will be specifically instructed to not penalize honesty concerning limitations.

3. **Theory Assumptions and Proofs**

   Question: For each theoretical result, does the paper provide the full set of assumptions and a complete (and correct) proof?

   Answer: [NA]

   Justification: The paper does not include theoretical results.

   Guidelines:
   - The answer NA means that the paper does not include theoretical results.
   - All the theorems, formulas, and proofs in the paper should be numbered and cross-referenced.
   - All assumptions should be clearly stated or referenced in the statement of any theorems.
   - The proofs can either appear in the main paper or the supplemental material, but if they appear in the supplemental material, the authors are encouraged to provide a short proof sketch to provide intuition.
   - Inversely, any informal proof provided in the core of the paper should be complemented by formal proofs provided in appendix or supplemental material.
   - Theorems and Lemmas that the proof relies upon should be properly referenced.

4. **Experimental Result Reproducibility**

   Question: Does the paper fully disclose all the information needed to reproduce the main experimental results of the paper to the extent that it affects the main claims and/or conclusions of the paper (regardless of whether the code and data are provided or not)?

   Answer: [Yes]

   Justification: We described the model structure as well as the training setup in section 5.2, and we will release the codes afterward.

   Guidelines:
   - The answer NA means that the paper does not include experiments.
   - If the paper includes experiments, a No answer to this question will not be perceived well by the reviewers: Making the paper reproducible is important, regardless of whether the code and data are provided or not.
   - If the contribution is a dataset and/or model, the authors should describe the steps taken to make their results reproducible or verifiable.
   - Depending on the contribution, reproducibility can be accomplished in various ways. For example, if the contribution is a novel architecture, describing the architecture fully might suffice, or if the contribution is a specific model and empirical evaluation, it may be necessary to either make it possible for others to replicate the model with the same dataset, or provide access to the model. In general. releasing code and data is often one good way to accomplish this, but reproducibility can also be provided via detailed descriptions for how to replicate the results, access to a hosted model (e.g., in the case of a large language model), releasing of a model checkpoint, or other means that are appropriate to the research performed.

- While NeurIPS does not require releasing code, the conference does require all submissions to provide some reasonable avenue for reproducibility, which may depend on the nature of the contribution. For example
  (a) If the contribution is primarily a new algorithm, the paper should make it clear how to reproduce that algorithm.
  (b) If the contribution is primarily a new model architecture, the paper should describe the architecture clearly and fully.
  (c) If the contribution is a new model (e.g., a large language model), then there should either be a way to access this model for reproducing the results or a way to reproduce the model (e.g., with an open-source dataset or instructions for how to construct the dataset).
  (d) We recognize that reproducibility may be tricky in some cases, in which case authors are welcome to describe the particular way they provide for reproducibility. In the case of closed-source models, it may be that access to the model is limited in some way (e.g., to registered users), but it should be possible for other researchers to have some path to reproducing or verifying the results.

5. **Open access to data and code**

   Question: Does the paper provide open access to the data and code, with sufficient instructions to faithfully reproduce the main experimental results, as described in supplemental material?

   Answer: [Yes]

   Justification: we will release the codes afterward.

   Guidelines:
   - The answer NA means that paper does not include experiments requiring code.
   - Please see the NeurIPS code and data submission guidelines (`https://nips.cc/public/guides/CodeSubmissionPolicy`) for more details.
   - While we encourage the release of code and data, we understand that this might not be possible, so "No" is an acceptable answer. Papers cannot be rejected simply for not including code, unless this is central to the contribution (e.g., for a new open-source benchmark).
   - The instructions should contain the exact command and environment needed to run to reproduce the results. See the NeurIPS code and data submission guidelines (`https://nips.cc/public/guides/CodeSubmissionPolicy`) for more details.
   - The authors should provide instructions on data access and preparation, including how to access the raw data, preprocessed data, intermediate data, and generated data, etc.
   - The authors should provide scripts to reproduce all experimental results for the new proposed method and baselines. If only a subset of experiments are reproducible, they should state which ones are omitted from the script and why.
   - At submission time, to preserve anonymity, the authors should release anonymized versions (if applicable).
   - Providing as much information as possible in supplemental material (appended to the paper) is recommended, but including URLs to data and code is permitted.

6. **Experimental Setting/Details**

   Question: Does the paper specify all the training and test details (e.g., data splits, hyperparameters, how they were chosen, type of optimizer, etc.) necessary to understand the results?

   Answer: [Yes]

   Justification: We mentioned this part in section 5.2 and in the appendix.

   Guidelines:
   - The answer NA means that the paper does not include experiments.
   - The experimental setting should be presented in the core of the paper to a level of detail that is necessary to appreciate the results and make sense of them.

- The full details can be provided either with the code, in appendix, or as supplemental material.

7. **Experiment Statistical Significance**

   Question: Does the paper report error bars suitably and correctly defined or other appropriate information about the statistical significance of the experiments?

   Answer: [No]

   Justification: error bars are not reported because it would be too computationally expensive.

   Guidelines:

   - The answer NA means that the paper does not include experiments.
   - The authors should answer "Yes" if the results are accompanied by error bars, confidence intervals, or statistical significance tests, at least for the experiments that support the main claims of the paper.
   - The factors of variability that the error bars are capturing should be clearly stated (for example, train/test split, initialization, random drawing of some parameter, or overall run with given experimental conditions).
   - The method for calculating the error bars should be explained (closed form formula, call to a library function, bootstrap, etc.)
   - The assumptions made should be given (e.g., Normally distributed errors).
   - It should be clear whether the error bar is the standard deviation or the standard error of the mean.
   - It is OK to report 1-sigma error bars, but one should state it. The authors should preferably report a 2-sigma error bar than state that they have a 96% CI, if the hypothesis of Normality of errors is not verified.
   - For asymmetric distributions, the authors should be careful not to show in tables or figures symmetric error bars that would yield results that are out of range (e.g. negative error rates).
   - If error bars are reported in tables or plots, The authors should explain in the text how they were calculated and reference the corresponding figures or tables in the text.

8. **Experiments Compute Resources**

   Question: For each experiment, does the paper provide sufficient information on the computer resources (type of compute workers, memory, time of execution) needed to reproduce the experiments?

   Answer: [Yes]

   Justification: We mentioned this part in section 5.2.

   Guidelines:

   - The answer NA means that the paper does not include experiments.
   - The paper should indicate the type of compute workers CPU or GPU, internal cluster, or cloud provider, including relevant memory and storage.
   - The paper should provide the amount of compute required for each of the individual experimental runs as well as estimate the total compute.
   - The paper should disclose whether the full research project required more compute than the experiments reported in the paper (e.g., preliminary or failed experiments that didn't make it into the paper).

9. **Code Of Ethics**

   Question: Does the research conducted in the paper conform, in every respect, with the NeurIPS Code of Ethics `https://neurips.cc/public/EthicsGuidelines`?

   Answer: [Yes]

   Justification: We don't have such a problem in our paper.

   Guidelines:

   - The answer NA means that the authors have not reviewed the NeurIPS Code of Ethics.

- If the authors answer No, they should explain the special circumstances that require a deviation from the Code of Ethics.
- The authors should make sure to preserve anonymity (e.g., if there is a special consideration due to laws or regulations in their jurisdiction).

10. **Broader Impacts**

Question: Does the paper discuss both potential positive societal impacts and negative societal impacts of the work performed?

Answer: [NA]

Justification: For now, this work does not have much social impact. Current work lies mainly in the simulation phase, which is still far from practical application.

Guidelines:

- The answer NA means that there is no societal impact of the work performed.
- If the authors answer NA or No, they should explain why their work has no societal impact or why the paper does not address societal impact.
- Examples of negative societal impacts include potential malicious or unintended uses (e.g., disinformation, generating fake profiles, surveillance), fairness considerations (e.g., deployment of technologies that could make decisions that unfairly impact specific groups), privacy considerations, and security considerations.
- The conference expects that many papers will be foundational research and not tied to particular applications, let alone deployments. However, if there is a direct path to any negative applications, the authors should point it out. For example, it is legitimate to point out that an improvement in the quality of generative models could be used to generate deepfakes for disinformation. On the other hand, it is not needed to point out that a generic algorithm for optimizing neural networks could enable people to train models that generate Deepfakes faster.
- The authors should consider possible harms that could arise when the technology is being used as intended and functioning correctly, harms that could arise when the technology is being used as intended but gives incorrect results, and harms following from (intentional or unintentional) misuse of the technology.
- If there are negative societal impacts, the authors could also discuss possible mitigation strategies (e.g., gated release of models, providing defenses in addition to attacks, mechanisms for monitoring misuse, mechanisms to monitor how a system learns from feedback over time, improving the efficiency and accessibility of ML).

11. **Safeguards**

Question: Does the paper describe safeguards that have been put in place for responsible release of data or models that have a high risk for misuse (e.g., pretrained language models, image generators, or scraped datasets)?

Answer: [NA]

Justification: The paper poses no such risks.

Guidelines:

- The answer NA means that the paper poses no such risks.
- Released models that have a high risk for misuse or dual-use should be released with necessary safeguards to allow for controlled use of the model, for example by requiring that users adhere to usage guidelines or restrictions to access the model or implementing safety filters.
- Datasets that have been scraped from the Internet could pose safety risks. The authors should describe how they avoided releasing unsafe images.
- We recognize that providing effective safeguards is challenging, and many papers do not require this, but we encourage authors to take this into account and make a best faith effort.

12. **Licenses for existing assets**

Question: Are the creators or original owners of assets (e.g., code, data, models), used in the paper, properly credited and are the license and terms of use explicitly mentioned and properly respected?

Answer: [Yes]

Justification: We cite the articles, models, and datasets mentioned in our paper.

Guidelines:

- The answer NA means that the paper does not use existing assets.
- The authors should cite the original paper that produced the code package or dataset.
- The authors should state which version of the asset is used and, if possible, include a URL.
- The name of the license (e.g., CC-BY 4.0) should be included for each asset.
- For scraped data from a particular source (e.g., website), the copyright and terms of service of that source should be provided.
- If assets are released, the license, copyright information, and terms of use in the package should be provided. For popular datasets, `paperswithcode.com/datasets` has curated licenses for some datasets. Their licensing guide can help determine the license of a dataset.
- For existing datasets that are re-packaged, both the original license and the license of the derived asset (if it has changed) should be provided.
- If this information is not available online, the authors are encouraged to reach out to the asset's creators.

13. **New Assets**

Question: Are new assets introduced in the paper well documented and is the documentation provided alongside the assets?

Answer: [NA]

Justification: The paper does not release new assets.

Guidelines:

- The answer NA means that the paper does not release new assets.
- Researchers should communicate the details of the dataset/code/model as part of their submissions via structured templates. This includes details about training, license, limitations, etc.
- The paper should discuss whether and how consent was obtained from people whose asset is used.
- At submission time, remember to anonymize your assets (if applicable). You can either create an anonymized URL or include an anonymized zip file.

14. **Crowdsourcing and Research with Human Subjects**

Question: For crowdsourcing experiments and research with human subjects, does the paper include the full text of instructions given to participants and screenshots, if applicable, as well as details about compensation (if any)?

Answer: [Yes]

Justification: We describe the experimental setup in section 5.4 and in the appendix.

Guidelines:

- The answer NA means that the paper does not involve crowdsourcing nor research with human subjects.
- Including this information in the supplemental material is fine, but if the main contribution of the paper involves human subjects, then as much detail as possible should be included in the main paper.
- According to the NeurIPS Code of Ethics, workers involved in data collection, curation, or other labor should be paid at least the minimum wage in the country of the data collector.

15. **Institutional Review Board (IRB) Approvals or Equivalent for Research with Human Subjects**

Question: Does the paper describe potential risks incurred by study participants, whether such risks were disclosed to the subjects, and whether Institutional Review Board (IRB) approvals (or an equivalent approval/review based on the requirements of your country or institution) were obtained?

Answer: [No]

Justification: The subjects in our experiment had no such risk.

Guidelines:

- The answer NA means that the paper does not involve crowdsourcing nor research with human subjects.
- Depending on the country in which research is conducted, IRB approval (or equivalent) may be required for any human subjects research. If you obtained IRB approval, you should clearly state this in the paper.
- We recognize that the procedures for this may vary significantly between institutions and locations, and we expect authors to adhere to the NeurIPS Code of Ethics and the guidelines for their institution.
- For initial submissions, do not include any information that would break anonymity (if applicable), such as the institution conducting the review.

