# OpenReview forum: "Language-Driven Interactive Traffic Trajectory Generation"
_NeurIPS.cc/2024/Conference — NeurIPS 2024 poster_

### Official Review · Reviewer_awjY · 2024-07-07

**Soundness:** 3
**Presentation:** 3
**Contribution:** 3
**Rating:** 5
**Confidence:** 4

**Summary:**

This paper proposes a large language model-based traffic trajectory generation method. Due to the designed interaction interpretation mechanism, the proposed method can generate a better corresponding trajectory. In addition, to improve the generation quality, the authors also proposed a well-designed prompt and a two-step feature aggregation. Finally, the authors also considered a variety of experiments to verify the effectiveness of the method.

**Strengths:**

i) The LLM-based traffic trajectory generator can generate interactive traffic trajectories.

ii) A well-designed interaction-aware prompt and code-to-trajectory decoder.

iii) In addition to quantitative analysis, a visual case study is also given in the experiment.

**Weaknesses:**

i) The author did not compare with the two related baseline methods.

ii) There are some collisions in the generated trajectory.

iii) The authors did not evaluate the diversity and scalability of the proposed generation method.

Minors:

Line 43: "a LLM-based" -> an

Line 631: “7 and 8 illustrate” -> “Figures 7 and 8”

**Questions:**

i) Since the training of the proposed method also still relies on pre-collected ground truth data, I think the authors need to compare it with CTG and CTG++. Another reason is these two methods are also based on language conditions.

ii) How diverse are the trajectories generated by the proposed method? Will they be similar to the training data?

iii) In some visualization results, there are some collisions in the generated vehicle trajectories, such as the red vehicle in the overtaking in Figure 5, and the orange vehicles in Figures 7(b) and 8(b). What is the reason for this?

iv) What is the size of the trajectory data generated by the proposed method? The visualization results show that the number of vehicles is not large. Can a more complex trajectory be generated?

**Limitations:**

Yes

---

> ### Author Rebuttal · Authors · 2024-08-06
>
> ##  **Response to Reviewer awjY**
> $\textbf{Question1:}$ Since the training of the proposed method also still relies on pre-collected ground truth data, I think the authors need to compare it with CTG and CTG++. Another reason is these two methods are also based on language conditions.
>
> $\textbf{Answer1:}$ The setting of CTG and CTG++ is different from our method setting of language-conditioned trajectory generation. CTG and CTG++ require past trajectory observations to generate traffic trajectories, as outlined in their problem formulation (Sec.III-A in CTG and Sec 3.1 in CTG++). Without past trajectory observations, these models are unable to produce plausible trajectories. This is the primary reason why these two models were not selected when comparing our model to the baseline models.
>
> $\textbf{Question2:}$ How diverse are the trajectories generated by the proposed method? Will they be similar to the training data?
>
> $\textbf{Answer2:}$  Because of the generalization ability and multiple possibilities of the LLM's response,  diversified scenarios can be generated from the same language input. Figure S4-1, S4-2 and S4-3 in the rebuttal pdf demonstrate one such case. For the same language input, the LLM will generate satisfying responses with variations, reflecting the realism and diversity of the generated scenarios by our model. Additionally, because of the LLM's generalization ability and multiple possibilities of LLM's response, the generated trajectories may differ significantly from the training data.
>
> $\textbf{Question3:}$ In some visualization results, there are some collisions in the generated vehicle trajectories, such as the red vehicle in the overtaking in Figure 5, and the orange vehicles in Figures 7(b) and 8(b). What is the reason for this?
>
> $\textbf{Answer3:}$ The vehicles involved in collisions in these figures mentioned are from scenarios generated by baseline models (LCTGen). This is due to two main reasons:  i) the baseline method does not consider vehicle interaction and generates vehicle trajectory independently, which can lead to conflicts or collisions between different vehicle trajectories. ii) the intermediate code design in the baseline method does not account for the global movement trend, thus there may be conflicts between the vehicle end state and map boundary. In comparison, our method i) considers interaction-aware numerical codes to jointly generate interaction-aware vehicle trajectories to avoid unreasonable collisions; ii) introduces the global trend in the vehicle code V to provide a more reasonable overall trajectory moving states, thereby avoid the collision between the vehicle end state and map boundary.
>
> $\textbf{Question4:}$ What is the size of the trajectory data generated by the proposed method? The visualization results show that the number of vehicles is not large. Can a more complex trajectory be generated?
>
> $\textbf{Answer4:}$ We generate a 5-second trajectory at 10 fps for each vehicle.  For the requirement of generating more complex traffic scenarios, LLM can smoothly assign appropriate vehicle codes to each vehicle and analyze inter-agent interactions to produce reasonable traffic scenarios. When the number of agents increases, our model is still able to generate scenarios that conform to linguistic descriptions. Figure S5 shows our model's ability to generate complex scenarios with more than 5 vehicles. Even as the number of vehicles in the scene increases, our model maintains its performance in scenario generation, handling cases with up to ten vehicles well, whereas the baseline method often results in significant vehicle collisions.

---

> ### Comment · Reviewer_awjY · 2024-08-10
>
> Thanks for your reply.

---

> > ### Author Response · Authors · 2024-08-11
> >
> > Thank you for your intresent. We would like to know whether you have any further questions or requests about our work, and we would appreciate it if you could kindly reconsider your review score, taking our rebuttal into account if we have addressed the primary concerns you mentioned in your review.

---

> > > ### Author Response · Authors · 2024-08-13
> > >
> > > We would like to know if you have any additional questions or requests regarding our work. If we have adequately addressed the primary concerns mentioned in the review, we  would appreciate it if you could consider revising your review score based on our rebuttal.

---

### Official Review · Reviewer_xSeP · 2024-07-09

**Soundness:** 3
**Presentation:** 3
**Contribution:** 3
**Rating:** 6
**Confidence:** 4

**Summary:**

This work proposes a novel framework for traffic trajectory generation with natural language description of interactions, called InteractTraj. Specifically, the proposed framework consists of two main components: a language-to-code encoder and a code-to-trajectory decoder. The language-to-code encoder is designed to generate the code representation of the natural language description, and the code-to-trajectory decoder is used to generate the trajectory based on the code representation. The proposed framework is evaluated on two dataset and shows promising results compared to the state-of-the-art methods.

**Strengths:**

1. This paper use language description as the conditional information for controlling the generation of traffic trajectories, which is a interesting idea.
2. This paper is well-written and easy to follow. Each section and each component of the model is clearly described.
3. The proposed framework is evaluated on two real-world datasets and shows promising results compared to the state-of-the-art methods. The authors also do a series of experiments like ablation study and user study to further validate the effectiveness of the proposed framework.

**Weaknesses:**

1. It seems the proposed framework is computationally expensive; the authors should provide the details about the efficiency of the model.
2. Is the language description encoder used in this paper GPT-4? There is no comparison of the effects of different large language models as the encoder.
3. Controllability is an important contribution of this paper, which should be validated more intuitively (visual) or quantitatively.

**Questions:**

1. Could the proposed model handle the different length of the generated trajectories? or it only generates the fixed-length trajectories?

**Limitations:**

This paper addresses the problem of using language for controllable trajectory generation and discusses the limitations of the model.

---

> ### Author Rebuttal · Authors · 2024-08-06
>
> ##  **Response to Reviewer xSeP**
>
> $\textbf{Question1:}$ It seems the proposed framework is computationally expensive; the authors should provide the details about the efficiency of the model.
>
> $\textbf{Answer1:}$  The overall resources needed for model training and inference are not expensive. Only the code-to-trajectory decoder needs to be trained as the LLM in our model encoder does not need training. It takes about 12 hours for 100 epochs on 4 NVIDIA GeForce RTX040 GPUs for the decoder training process. During the inference phase, it takes about 30 seconds to generate a scenario,  which includes about mainly 30 seconds to receive the response from the LLM and about 0.2 seconds to generate corresponding trajectories.
>
>
> $\textbf{Question2:}$ Is the language description encoder used in this paper GPT-4? There is no comparison of the effects of different large language models as the encoder.
>
> $\textbf{Answer2:}$ We use GPT-4 as the language description encoder in our paper, but many other LLMs also can be incorporated into our work. Here we further show the result of applying two other representative LLMs as encoders, including the Llama3-3.1-70b and Mistral-large. See Figure S2, which illustrates the visualization results provided by both Llama3 and Mistral-large for different types of interaction. We see that applying other LLMs also achieves compliant scenario generation.
>
>
> $\textbf{Question3: }$ Controllability is an important contribution of this paper, which should be validated more intuitively (visual) or quantitatively.
>
> $\textbf{Answer3:}$  Our method can control the specified vehicle trajectory generation in the scene. As controllability relies on the language input, here we validated more intuitively through visualization. Figure S3 shows the generation results under four different control descriptions applied to ego vehicles.  We observe that our method can control specific vehicle trajectories through different linguistic descriptions.
>
>
> $\textbf{Question4:}$ Could the proposed model handle the different lengths of the generated trajectories? or it only generates the fixed-length trajectories?
>
> $\textbf{Answer4:}$ The length of the generated trajectories is pre-defined before the model is trained. In our paper, we generate 5-second trajectories at 10 fps in experiments. The setting of fixed-length trajectories is common in related tasks like trajectory generation [1,2] and prediction [3,4].
> [1] Learning to generate diverse and realistic traffic scenarios, ICRA 2023
> [2] Language-conditioned traffic generation, CoRL 2023
> [3] Motiondiffuser: Controllable multi-agent motion prediction using diffusion, CVPR 2023
> [4] MotionLM: Multi-Agent Motion Forecasting as Language Modeling, CVPR 2023

---

> > ### Comment · Reviewer_xSeP · 2024-08-13
> >
> > I appreciate the author's response and the extra work. This addresses some of my concerns. Currently, I think the scores given are reasonable and appropriate.

---

### Official Review · Reviewer_YJzW · 2024-07-12

**Soundness:** 3
**Presentation:** 3
**Contribution:** 4
**Rating:** 7
**Confidence:** 3

**Summary:**

The paper introduces a novel method called InteractTraj, which is the first language-driven traffic trajectory generator capable of producing interactive traffic trajectories. This method is critical for advancing autonomous driving technology because it can generate realistic and controllable vehicle interaction trajectories based on natural language instructions, addressing the limitations of previous approaches that focused solely on generating trajectories for individual traffic participants without considering complex traffic dynamics.

InteractTraj features an innovative language-to-code encoder with an interaction-aware encoding strategy that translates abstract trajectory descriptions into concrete, interaction-aware numerical codes. Additionally, it includes a code-to-trajectory decoder that employs interaction-aware feature aggregation, integrating vehicle interactions, environmental map data, and vehicle movements to generate interactive traffic trajectories.

Experimental results show that InteractTraj outperforms SOTA methods in generating interactive traffic trajectories under diverse natural language commands, offering more realistic simulations of interactive traffic scenarios. This research is significant for driving simulation, particularly in reducing the costs of safety-critical scenarios by recreating real-world situations in virtual environments, thereby accelerating the development of autonomous driving technologies.

**Strengths:**

**Originality**

The paper introduces a significant innovation in the field of autonomous vehicle technology and driving simulation by proposing InteractTraj, the first language-driven traffic trajectory generator capable of producing interactive traffic trajectories. This concept stands out due to its novelty in interpreting abstract trajectory descriptions into concrete, interaction-aware numerical codes, which then guide the generation of realistic traffic scenarios. The method's originality is further highlighted by its ability to encapsulate complex interactive dynamics that were previously unaddressed by focusing solely on individual traffic participant trajectories.

**Quality**

The research demonstrates high-quality methodology and rigorous experimental validation. The authors propose a language-to-code encoder with an interaction-aware encoding strategy, followed by a code-to-trajectory decoder that synergizes vehicle interactions with the environment. The comprehensive experiments conducted show superior performance over state-of-the-art (SoTA) methods, substantiated by quantitative metrics and qualitative assessments through user studies. The results indicate that InteractTraj not only generates more realistic and controllable interactive traffic trajectories but also effectively captures the essence of diverse natural language commands.

**Clarity**

The paper is well-structured and clearly articulated, making it accessible to reseachers in the field. The abstract succinctly summarizes the contributions and the significance of the work. The introduction sets the context by highlighting the importance of driving simulations in autonomous vehicle development and identifies the gap addressed by the proposed solution. The methodology is explained with sufficient detail to allow for replication, supported by figures and tables that aid in understanding the findings.

**Significance**

The impact of this work is substantial, particularly in the realm of autonomous driving and traffic simulation. By enabling the generation of interactive traffic scenarios through natural language commands, InteractTraj opens up new possibilities for creating diverse and realistic driving conditions, which is critical for testing and validating autonomous vehicle algorithms. This could potentially lead to safer and more reliable autonomous systems by exposing them to a wide range of driving situations that might be difficult or dangerous to replicate in real life.

In summary, the paper excels in its original contribution to the field, the high quality of its research, the clarity of its presentation, and the significant implications of its findings for autonomous vehicle technology and driving simulation.

**Weaknesses:**

Despite the significant contributions and innovative approach outlined in the paper, there is a weaknesses where the work could be improved to better achieve its stated goals:

1.**Robustness to Ambiguous or Complex Language Descriptions**: The paper does not extensively explore how the system handles ambiguous or complex linguistic instructions. Since natural language can be nuanced and subject to interpretation, evaluating the system's ability to generate accurate trajectories when faced with unclear or highly detailed descriptions would provide a clearer picture of its robustness and flexibility.

**Questions:**

1. **Scope of Interaction Types**: The study focuses on a set of predefined interaction types. Could the authors clarify how the model performs when dealing with emergent or less common interaction types that might not have been explicitly trained on, like parallel driving or platooning? Would the model generalize well to such scenarios, or would it require retraining?

2. **Language Ambiguity Handling**: How does InteractTraj handle ambiguity in natural language descriptions? For instance, phrases like "the car behind" can be ambiguous in dense traffic scenarios. Does the model prioritize proximity or other factors when resolving such ambiguities?

**Limitations:**

The authors acknowledge some limitations of their work and outline future directions for addressing these constraints. However, there is room for improvement in terms of comprehensively addressing the limitations and discussing potential negative societal impacts. Here are some areas that could benefit from further exploration and discussion:

1. **Limited Scope of Traffic Participants**: Currently, the focus is on generating trajectories for vehicles only. The system does not account for other traffic participants such as pedestrians, cyclists, or other non-motorized vehicles. This limitation could be addressed by extending the model to encompass a broader range of entities in the traffic ecosystem.

2. **Map Generation Constraints**: The authors admit that map generation is restricted by the available map library. To enhance flexibility and realism, the system could be expanded to generate more diverse and flexible maps, possibly through generative models that create new environments based on learned patterns.

3. **Complex Interaction Handling**: While the model performs well with common interaction types, its ability to handle more complex or rare interactions has not been thoroughly evaluated. The authors could consider testing the model's performance in scenarios involving intricate multi-agent interactions to understand its limitations better.

---

> ### Author Rebuttal · Authors · 2024-08-06
>
> ## **Response to Reviewer YJzW**
>
> $\textbf{Question1:}$  Clarify how the model performs when dealing with emergent or less common interaction types, like parallel driving or platooning? Would the model generalize well to such scenarios, or would it require retraining?
>
> $\textbf{Answer1:}$ Our model can generate compliant scenarios when dealing with less common interaction types. The left part of Figure S1 in the rebuttal pdf shows the experimental results. We present the results for rare interaction types including mentioned uncommon cases of parallel driving (Figure S1-1), platooning (Figure S1-2), and another uncommon case involving pulling over (Figure S1-3). We see that for less common interaction types, our method effectively translates the relevant behaviors to generate compliant scenarios.
> Our model is able to generalize scenarios with emergent or less common interactions without requiring retraining. The main reason is that the LLM used in our encoding process possesses strong generalization abilities. The large language model can understand linguistic descriptions of uncommon interaction scenarios and convert them into appropriate numerical codes for decoding. Then the decoding process, which also has generalization ability due to training with massive numerical codes, would translate these codes into trajectories and generate compliant scenarios.
>
> $\textbf{Question2:}$ How does InteractTraj handle ambiguity in language descriptions, like phrases like "the car behind" in dense traffic scenarios? Does the model prioritize proximity or other factors when resolving such ambiguities?
>
> $\textbf{Answer2:}$ We leverage the reasoning ability of the large language model to manipulate linguistic inputs to solve possible ambiguity problems. We do not introduce additional prioritized proximity or other factors. Ambiguous situations mainly fall into two categories: those where the reference to the object is unclear, and those where there is a contradiction within the language instruction itself.
> - For the first type, LLM would understand language descriptions and convert them into numerical codes to satisfy one of the meanings of phrases. We present the result of the mentioned phrase "the car behind", see the right part of Figure S1 in the rebuttal pdf. We see that when the reference to "the car behind" is ambiguous (Figure S1-4), the LLM would randomly designate one vehicle as the "behind car". However, with clearer descriptions like "the car behind ego car" (Figure S1-5), the LLM would accurately handle the descriptions.
> - For the second type involving self-contradictory requirements, the scenarios generated would partially meet the instructions' criteria. Figure S1-6 illustrates this with an example of language instruction "A car is turning left and moving in a straight line." LLM may opt to generate an intersection in the scene to fulfill the left-turning requirement while disregarding the requirement for straight-line movement.
> To better tackle language ambiguity, an optimal solution involves introducing LLM-human interaction to iteratively verify language descriptions. This approach will be explored in our future work.
>
> $\textbf{Question3:}$ Currently, the focus is on generating trajectories for vehicles only. This limitation could be addressed by extending the model to encompass a broader range of entities.
>
> $\textbf{Answer3:}$ At present, our work concentrates on trajectory generation for vehicles since vehicles are the most common subject, specifically creating interactive scenarios by modeling vehicle interactions.  To incorporate various types of traffic participants, such as pedestrians and cyclists, we could utilize specialized heterogeneous networks as the backbone for the decoder network. Additionally, we plan to design specialized dynamic models, such as the bicycle kinematic model, tailored for different subjects. This comprehensive systematic design will be pursued in future work.
>
> $\textbf{Question4:}$ The authors admit that map generation is restricted by the available map library. The system could be expanded to generate more diverse and flexible maps, possibly through generative models.
>
> $\textbf{Answer4:}$ We use the current map library method since i) it ensures the obtained map is totally realistic and ii) the built map library could already support plenty of trajectory generation cases.  Although a map generation method could generate more flexible maps, it may encounter the problem of being unrealistic. Meanwhile, in this work, our main solving problem is how to generate traffic trajectories with interactions. Thus, we hope the map is ensured to be realistic to better evaluate our trajectory generation design. In future work, we will switch the focus to map generation. To expand our method with more efficient map generation methods to obtain maps that conform more closely to linguistic inputs, we plan to use masked prediction for map expansion given partial map observations or use diffusion models for map generation.
>
> $\textbf{Question5:}$ While the model performs well with common interaction types, its ability to handle more complex or rare interactions has not been thoroughly evaluated. The authors could consider testing the model's performance in scenarios involving intricate multi-agent interactions.
>
> $\textbf{Answer5:}$ We have analyzed and shown some qualitative visualizations of complex or rare interaction cases in the responses A1 and A2. We will include more results of complex and rare interactions in the revised version. The quantitative evaluation of intricate multi-agent interactions is restricted by the existing benchmarks since there are no direct datasets full of intricate interactions labeled with language descriptions. In the future, when direct language-to-scenario datasets or specific datasets focusing on intricate interactions are available, we will conduct a more thorough evaluation of intricate multi-agent interaction cases.

---

> ### Comment · Reviewer_YJzW · 2024-08-12
>
> Thank you for taking the time to address my comments

---

### Author Rebuttal · Authors · 2024-08-06

Thank you for your valuable feedback!
We have responded to questions based on the opinions of each reviewer. The attached PDF contains various figures. Please review it.

---

### Decision · Program_Chairs · 2024-09-25

**Decision:**

Accept (poster)

**Comment:**

This paper introduces InteractTraj, a novel language-driven traffic trajectory generator capable of producing interactive traffic trajectories. The method uses a language-to-code encoder with an interaction-aware encoding strategy and a code-to-trajectory decoder employing interaction-aware feature aggregation. The authors evaluate their approach on two real-world datasets and compare it with state-of-the-art methods.

Reviewers highlight several strengths of the paper:
1. Originality: InteractTraj is noted as the first of its kind in addressing interactive traffic trajectory generation from natural language descriptions.
2. Methodology: The paper presents a well-structured and clearly articulated methodology.
3. Experimental Validation: Comprehensive experiments include quantitative metrics, qualitative assessments, and user studies.
4. Potential Impact: The work shows significant implications for autonomous vehicle and driving simulation.

Overall, reviewers view the paper positively, with two recommending acceptance and one giving a borderline accept.

Recommendation:

Based on the reviews, the recommendation is to accept the paper. The proposed approach and potential impact outweigh the noted limitations. However, the authors should address the following in the camera-ready version:
1. Include comparisons with language-based baseline methods (CTG and CTG++).
2. Provide analysis of the model's efficiency and computational requirements.
3. Address collision issues in generated trajectories.
4. Expand on the model's ability to handle diverse and scalable scenarios.
5. Discuss how the model handles ambiguous language descriptions and complex interaction types.